# Delivering equity in low-carbon multisector infrastructure planning

Adil Ashraf ⓘ [1], Mohammed Basheer ⓘ [1,2,3], Jose M. Gonzalez ⓘ [1], Eduardo A. Martínez Ceseña [4,5], Mikiyas Etichia ⓘ [1], Emmanuel Obuobie ⓘ [6], Andrea Bottacin-Busolin ⓘ [1,7], Jan Adamowski [8], Mathaios Panteli ⓘ [4,9] & Julien J. Harou ⓘ [1,10] ✉

Many countries worldwide are transitioning from fossil fuel-dependent economies to carbon neutrality, driven by the 2030 agenda for sustainable development and the Paris Agreement. However, without considering the regional distribution of essential services like water and energy, this transition could inadvertently maintain or increase inequities, threatening sustainable development. Here, we argue that spatial equity of benefits should be considered in planning low-carbon energy transitions, especially in developing countries with multisector interdependencies and high service disparities between regions. We propose an analytical framework that can help analysts and policymakers plan for regionally equitable climate-compatible futures. The multisector design framework combines integrated river basin-power system simulation with artificial intelligence design tools. The utility of the framework is demonstrated for Ghana by identifying the most efficient infrastructure intervention portfolios and their implied trade-offs between spatial equity in water and energy service provision, carbon emissions, food production, and river ecosystem performance. Case-study results show that an equitable low-carbon energy transition will require increased investments in renewable energy and transmission alongside more informed infrastructure system planning. With low renewable investments, equity can be improved, but at the cost of higher emissions and electricity supply curtailments.

Reducing socio-economic inequalities is integral to achieving the Sustainable Development Goals (SDGs)[1,2]. For instance, SDG 10 aims to reduce inequalities within and among countries[1], and other goals stress equitable development globally[1,2]. However, more than 70% of the global population is experiencing growing socio-economic inequalities[3], hindering progress towards sustainable development. Nearly 650 million people in the world still live in extreme poverty[4], and around 30% of the global population is food insecure[5]. Climate change, the COVID-19 pandemic, armed conflicts, and the rising cost of living disproportionately affect the world's most vulnerable people, who often have limited or no access to essential services[3,4,6].

[1]Department of Civil Engineering and Management, The University of Manchester, Manchester, UK. [2]Department of Civil and Mineral Engineering, University of Toronto, Toronto, Ontario, Canada. [3]Thaer-Institute of Agricultural and Horticultural Sciences, Humboldt University of Berlin, Berlin, Germany. [4]Department of Electrical and Electronic Engineering, The University of Manchester, Manchester, UK. [5]Tyndall Centre for Climate Change Research, The University of Manchester, Manchester, UK. [6]Water Research Institute, Council for Scientific and Industrial Research, Accra, Ghana. [7]Department of Industrial Engineering, University of Padua, Padua, Italy. [8]Department of Bioresource Engineering, McGill University, Montreal, Quebec, Canada. [9]Department of Electrical and Computer Engineering, University of Cyprus, Nicosia, Cyprus. [10]Department of Civil, Environmental and Geomatic Engineering, University College London, London, UK. ✉e-mail: julien.harou@manchester.ac.uk

Several countries are transitioning from fossil fuel-driven economies to carbon neutrality to deliver on the SDGs[1] and the Paris Agreement[7], intending to limit global mean temperature rise to 2 degrees Celsius above pre-industrial levels. However, meeting the SDGs and mitigating climate change without considering broader socio-economic implications of interventions could exacerbate inequalities[8]. For instance, climate mitigation actions are expected to yield long-term benefits across multiple SDGs[9,10] but might have short-term trade-offs with other goals, like poverty reduction and inequality[10,11]. Inadequate planning for energy transitions could increase inequalities[12–17], for example in access to water and electricity. Realising the SDGs is challenged by the complexity of involving multiple disciplines, sectors, and actors in decision-making[18,19]. Policy decisions aimed at achieving individual SDGs could create trade-offs with other SDGs[9], and thus, policymakers should aim for an energy transition that creates synergies and balances trade-offs between societal goals[20,21].

Increased access to affordable, reliable, and sustainable energy is required to achieve the SDGs[1,20,22]. Electrification improves the quality of life and enables human development through better-equipped facilities for healthcare, education, and business[17,22,23]. However, over 700 million people globally do not have access to electricity, of whom more than 75% live in sub-Saharan Africa[4,24]. A renewables-based energy transition can help increase access to affordable and clean energy to meet the fast-growing energy demand and mitigate climate change[4,24–26]. Most countries included renewable energy targets in their Nationally Determined Contributions (NDCs)[27–29]. The renewable share of electricity generation rose by almost 8% between 2011 and 2021, and reached an installed capacity of 3146 gigawatts (GW), which accounts for 28% of the world's energy generation[27]. However, recent reports by the International Renewable Energy Agency (IRENA) and REN21 suggest that the share of renewables in electricity generation must increase to more than 60% to achieve energy and climate goals by 2030[26,27].

Energy transitions can produce regional disparities in renewable energy access and job creation in developed countries[12–15,30]. Strategic regional planning can help avoid these inequalities while simultaneously reducing greenhouse gas emissions. Many developing countries (those eligible for Official Development Assistance under the Development Assistance Committee (DAC) list of the Organisation for Economic Co-operation and Development (OECD)[31]) currently have high levels of disparities in (water and electricity) service provision. Planning low-carbon energy transitions in developing countries offers an opportunity to reduce these inequalities. The aspiration to consider equity in energy transitions is reflected in the recovery packages for recent global crises such as the COVID-19 pandemic, rising energy costs, and inflation[28,32]. The African Union Green Recovery Action Plan aims to build more equitable, greener, and sustainable economies[33]. The USA Inflation Reduction Act[34] includes a US$ (US dollars) 370 billion fund for renewable energy, climate crisis, and equity, and the Build Back Better plan[35,36] aims to invest US$ 2 trillion in green and inclusive growth, including US$ 400 billion for clean energy and 40% of investment benefits for communities that are marginalised and underserved by infrastructure services[37].

Energy systems are typically embedded within complex water-energy-food-ecosystem (WEFE) systems, in which actions in one part of the system impact the others. Jointly managing these resources allows consideration of their cross-sectoral interdependencies, trade-offs, and synergies, and subsequently, the development of more sustainable infrastructure development pathways. So far, research has assessed distributional equity in the context of individual sectors. For instance, in the power sector, power system models have been used to assess the trade-offs between least-cost and regionally equitable allocation of solar and wind power plants in Germany[15] and that of decentralised renewable energy sources in

Switzerland[14]. Others considered electrification inequality in planning national power systems in sub-Saharan Africa[16,17]. A few studies evaluated inequality in access to water resources[38–40]. However, despite frequently high levels of co-dependence between resource systems, to our knowledge, equity has not been considered in designing multisector infrastructure systems to reduce regional benefit distribution inequities and carbon emissions while improving overall system performance and alleviating inter-sectoral conflicts across multiple resource systems.

In this paper, we argue for considering spatial equity of water and energy services in planning low-carbon energy transitions, especially in developing countries with multisector linkages and service disparities between regions. We propose an analytical framework that can help analysts and policymakers plan national-scale equitable climate-compatible futures. The framework combines interlinked water resource and power systems simulators with artificial intelligence design tools. We demonstrate the framework using a case study of planning Ghana's future water-energy system for 2030-2040, identifying infrastructure intervention portfolios with efficient trade-offs between spatial equity in water and energy service provision, carbon emissions, food production, and river ecosystem performance.

## Results
### Equity as a policy goal in low-carbon multisector infrastructure planning

Classically, energy infrastructure planning considers goals such as minimising electricity supply curtailment, capital investment, and carbon emissions[41,42]. Other approaches extend the goals to include water resources and ecosystems, such as maximising hydropower generation and minimising damage to river ecosystems[43,44]. While such methods can design infrastructure plans that improve system performance at an aggregate level, equity in the regional distribution of water and energy services has not been considered to our knowledge. In this paper, we build on a water resource and power system simulation and design framework introduced by ref. 43 and extend it to include spatial equity of water and energy services as infrastructure planning goals. The resulting framework (Fig. 1) has three components: (1) identifying low-carbon energy transition goals for infrastructure planning with equity and carbon emissions as key drivers, and selecting infrastructure assets and resource allocation policies; (2) integrated water-energy system simulation coupled with artificial intelligence-assisted multi-objective planning to design equitable energy transition intervention portfolios; and (3) inclusive screening and deliberation of the best set of plans assisted by machine learning.

In the first component of the design framework, equity in regional benefit distribution is added as a policy objective of infrastructure planning, alongside other goals. We use the Gini index to quantify and minimise water and electricity access inequities. The Gini index is typically used to measure inequalities – ranging from zero (perfect equality) to one (total inequality). Our application of the index is designed to account for equity. For water supply, we apply the Gini index to water supplied relative to demand. For electricity supply, since some areas have lower electricity demand due to regional disparities in economic development, if the ratio of supply to demand were used, it would easily be satisfied. Instead, we apply the Gini index to electricity supplied per capita. While this does not account for inevitable variations in regional economic activities and lifestyles, it makes the most of available model outputs and allows incorporating a spatial measure of energy service equity into the framework. Intervention options that can be used to achieve the identified equity, environmental, and system performance goals are also selected in the first component of the framework. These interventions include infrastructure assets and resource allocation policies. The selected interventions are used in the second component of the framework to

generate a set of efficient multisector infrastructure and operation portfolios using a simulation-based artificial intelligence search design process.

In the second component of the framework, an integrated river basin-power system simulator is used to represent spatial WEFE nexus dynamics[43]. The river system is modelled using a water resource system model[45,46]. The river system model represents water resources infrastructure (e.g., dams) using a network structure driven by water supplies (hydrological inflows) and demands (e.g., municipal and irrigation water requirements and hydropower) and system operating rules. The power system is modelled using a direct current optimal power flow model[47]. The power system model simulates network connectivity, different power generation types including renewable energy generators, and high time resolution electricity demand profiles. The integration of the simulators uses a multi-actor object-oriented simulation framework[48]. This framework integrates the simulators at the model run timestep and coordinates their inputs and outputs to form a single simulation. The integrated WEFE infrastructure system simulator simulates managed rivers (with river inflows, existing and planned water storage reservoirs, hydropower, municipal water supply abstractions, irrigation, and flood recession agriculture) and energy systems (solar power, combined solar and storage, wind power, bioenergy, thermal power generators, electricity transmission lines, and electricity demands). The multisector simulator is connected to an artificial intelligence-assisted multi-objective evolutionary algorithm (MOEA) to identify efficient intervention portfolios[43].

In the third component of the framework, machine learning models, trained based on the search outputs (i.e., values of decision variables and objectives), are combined with the Shapley additive explanations (SHAP)[49] to determine the influence of intervention decisions on equity, environmental, and system performance indicators. The SHAP analysis of intervention options helps stakeholders explore how changing infrastructure and resource allocation policy options influences intervention portfolios. Further details on the analytical framework are given in the Methods.

## Equitable low-carbon energy transition in an African case

We use Ghana as a case study to demonstrate the importance of considering the spatial equity of water and energy services in designing equitable low-carbon futures. We use the framework above to design efficient power system intervention portfolios, including bioenergy, intermittent renewables (solar and wind), solar storage, and transmission lines, using projected electricity demands for 2030-2040. Figure 2 shows the regions of Ghana defined based on equally sized population groups. These regions are used to quantify spatial equity in water and energy access and to locate the energy infrastructure expansion options currently listed in Ghana's Power System Master Plan[50]. The electricity mix of Ghana comes from hydro (47%), gas (30%), and oil (23%)[51]. The installed capacity for existing hydropower is 1,580 megawatts (MW). The Pwalugu multi-purpose dam is under construction and will add an electricity generation capacity of 59 MW. The country's total thermal power installed capacity is 4325 MW[50]. Ghana had an electricity access rate of around 85% in 2021[24]. However, some challenges still exist, including power outages[52] (called Dumsor in Ghana, which means "off and on"), low electricity access rates in the Northeast and Northwest regions[50], high electrical transmission losses (approximately 20%[50]), and a high per capita energy-related carbon emission rate compared with other sub-Saharan African countries[51]. The government of Ghana aims to increase renewables in the power generation mix to 1363 MW[53] and achieve a 45% decrease in greenhouse gas emissions by 2030 compared to business-as-usual levels[54,55]. Supplementary Fig. 1 shows Ghana's national power system transmission network and existing and planned water-energy infrastructure. Supplementary Table 1 provides the type and number of nodes in the Ghana multisector infrastructure system model.

The design formulation for Ghana includes minimising the electricity access Gini index, carbon emissions from generation, electricity supply curtailment, irrigation water supply Gini index, power system capital and operating costs, and maximising agriculture yields and flood recession agriculture benefits (eight objectives). The multi-objective search optimises 170 interventions (decisions variables), including infrastructure expansion (solar, wind, combined solar and

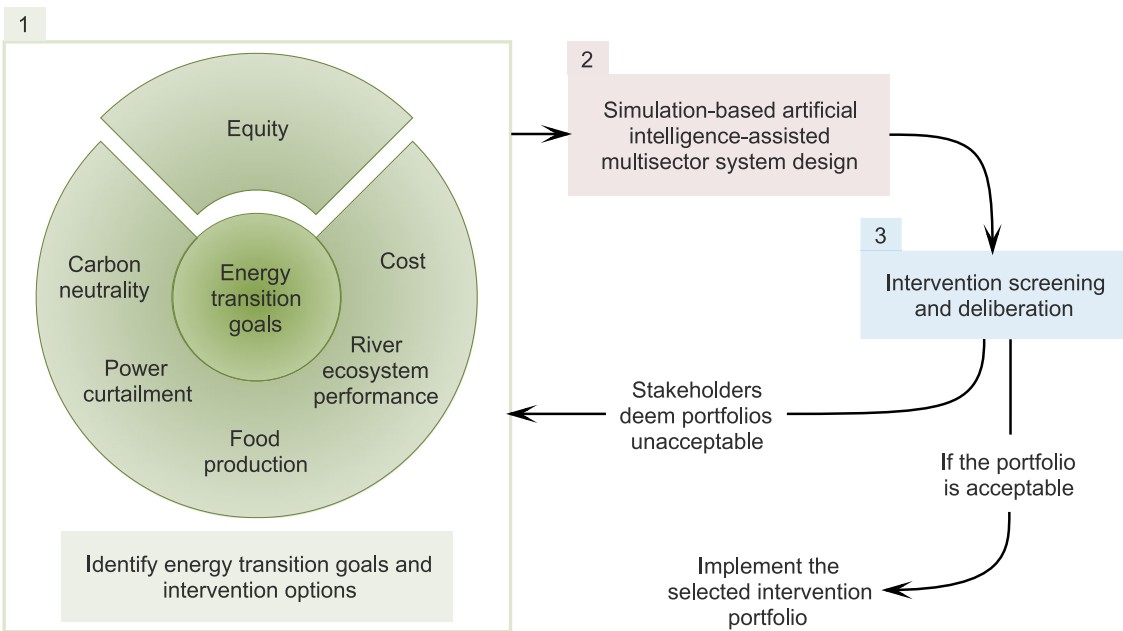

**Fig. 1 | Analytical framework for equitable low-carbon energy transitions in multisector water-energy systems.** The framework includes three components: (1) identifying low-carbon energy transition goals for infrastructure planning with equity and carbon emissions as key drivers, and selecting infrastructure assets and resource allocation policies; (2) integrated water-energy system simulation coupled with artificial intelligence-assisted multi-objective search for equitable energy transition intervention portfolios; and (3) inclusive screening and deliberation of the best set of plans assisted by machine learning.

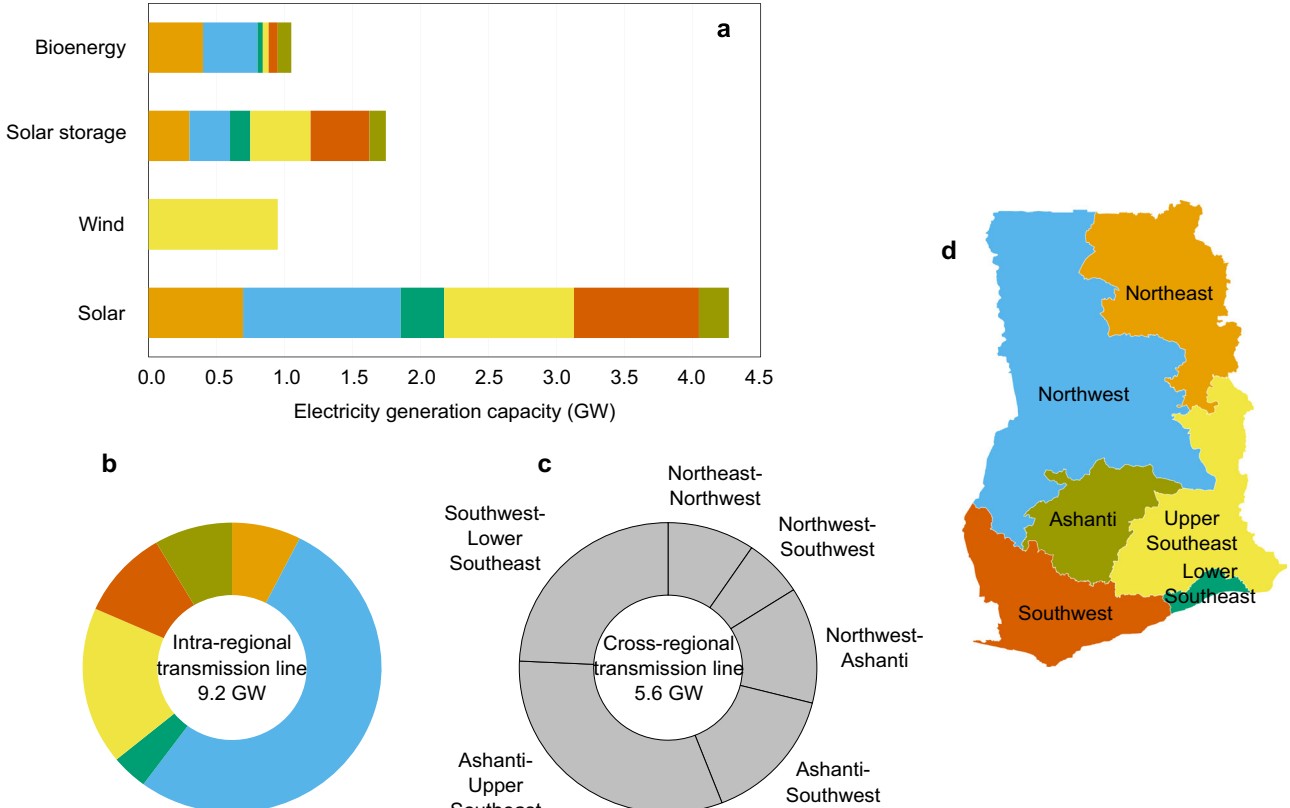

**Fig. 2 | Ghana's power system investment options considered in the proposed equity-informed design process. a–c** Ghana's power system infrastructure expansion is considered in the design process. **d** The regions of Ghana are defined based on equally sized populations (to quantify the electricity access Gini index). The irrigation and municipal water supply Gini indices are calculated between the northern (Northeast, Northwest, and Ashanti) and southern regions (Lower Southeast, Upper Southeast, and Southwest) due to the limited spatial data of nodes of irrigation and municipal water demands. The bar and segment colours in (**a**) and (**b**) correspond to the regions with the same colours in (**d**). GW stands for gigawatts. The map in (**d**) is based on data from Ghana Statistical Services (GSS). Source data are provided as a Source Data file via Zenodo at https://doi.org/10.5281/zenodo.14851474 (ref. 81).

storage, bioenergy, and transmission lines), operating rules for Akosombo, Bui, and Pwalugu reservoirs, and electricity and irrigation water allocation policies.

## Efficient and equitable low-carbon intervention portfolios

Figure 3a shows a parallel coordinates plot of the efficient infrastructure and operation portfolios produced for Ghana using the framework. Selected portfolios are indicated as thick coloured lines. The "baseline" portfolio (black line) reflects the performance of the infrastructure expansion proposed in the Ghana Power System Master Plan[50]. The "baseline" has high values of Gini coefficients for energy and irrigation water, carbon emissions, and load curtailment compared to the selected portfolios except for the "low energy Gini and high emissions" portfolio (magenta line). The selected portfolios (intervention bundles) show varying trade-offs between performance indicators. For instance, achieving low-carbon emissions (teal line) requires large investments in renewables and transmission lines, whereas less investment increases emissions (blue line). There is also a trade-off between power load curtailment and capital investment (teal and magenta lines). The "low energy Gini" portfolios (magenta and brown lines) improve equity by 7% compared to the "baseline" but alter emissions and curtailment depending on the level of investment. A reduction of US\$ 180 million in investment compared to the "baseline" portfolio increases emissions by 7% (magenta line), whereas an increase of US\$ 3,000 million results in a 5% reduction in emissions (brown line). Some Ghanaian regions have low use of electricity (e.g., Northwest and Northeast), resulting in a low electricity Gini index limit of 0.473. The "low emissions" portfolio (teal line) shows a 6% reduction in inequities in electricity access, a 25% reduction in carbon emissions, and a 1% increase in agricultural yield compared to the "baseline" while decreasing the load curtailment and irrigation water supply Gini index to zero, with an increase of US\$ 7240 million investment in renewables and transmission lines. Except for a 9% increase in carbon emissions, the "low curtailment" portfolio (blue line) is comparable to the "low emissions" portfolio (teal line), with an increase of US\$ 5280 million in investment compared to the "baseline".

Figure 3b shows the distribution of infrastructure expansion (i.e., how much new intermittent renewables and solar storage, bioenergy, and transmission lines) in the selected portfolios of Fig. 3a, c, d show carbon emissions of thermal and bioenergy power plants of the Fig. 3a portfolios. The "low emissions" and "low curtailment" portfolios include major investments in renewables (Fig. 3b). These portfolios improve system performance and decrease operating costs (by nearly 20%) compared to the "baseline". The "low emissions" portfolio results in a high capacity of new transmission lines, whereas transmission line expansion is lower in the "baseline" and "low energy Gini" portfolios. The renewables are lower in the "low energy Gini" portfolios. Figure 3c, d show that thermal power plants are the major contributors to carbon emissions; the power system continues to partially rely on them to meet the growing electricity demand.

Results show how changes in electricity access equity are impacted by the spatial distribution of investments (i.e., prioritising underserved regions and moving resources from more developed to least developed regions to ensure equitable regional coverage of services - Fig. 4) and indicate the need for large investments in renewables to

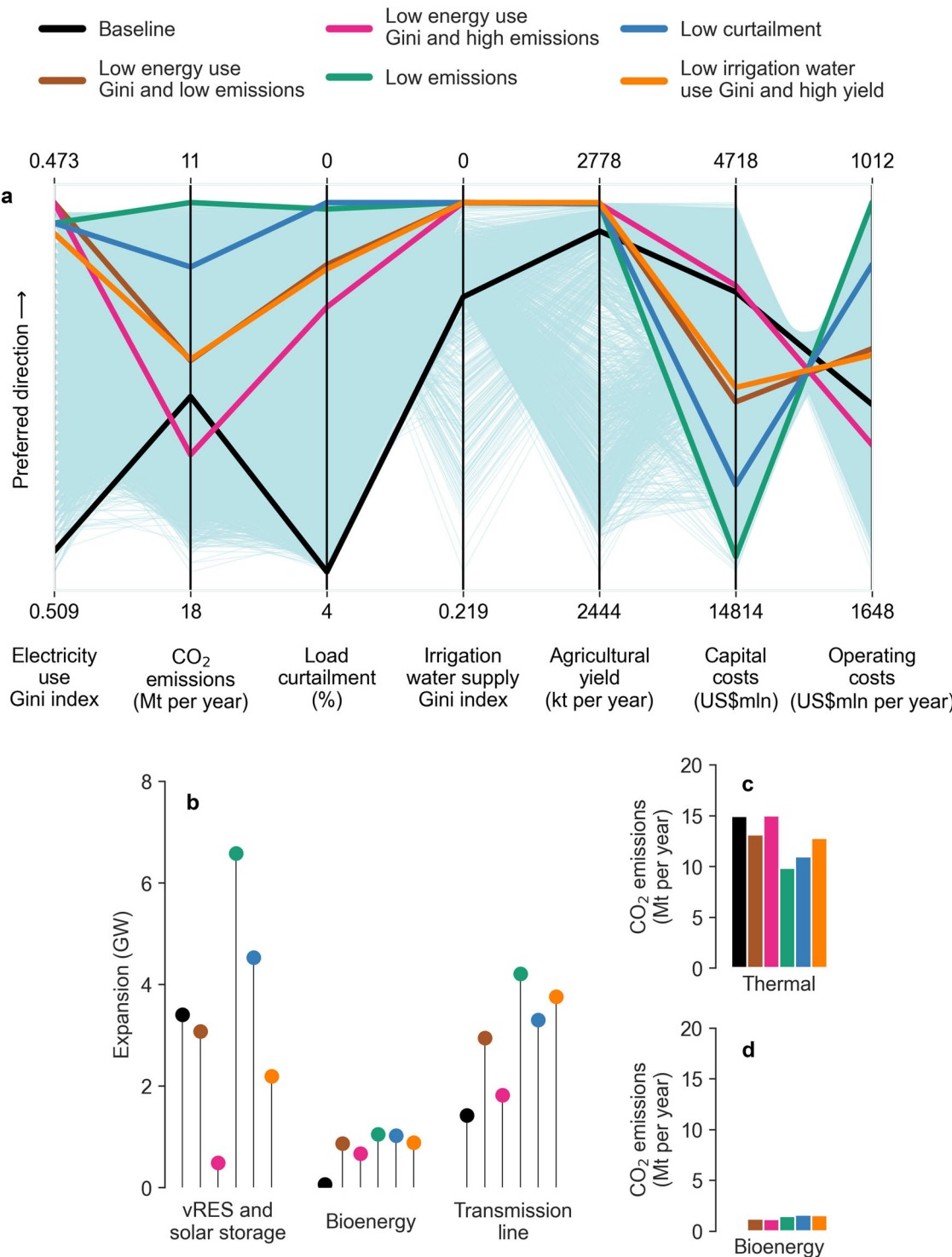

**Fig. 3 | Trade-offs between equity, river basin and power system performance indicators. a** Parallel coordinates plot of the baseline and best-performing (Pareto-efficient) infrastructure and operation portfolios, **b** Aggregate portfolios of infrastructure expansion, and **c**, **d** Carbon emissions from thermal and bioenergy power plants. Each line in (**a**) depicts the performance achieved by one of the Pareto-efficient portfolios (except the "baseline"); thick coloured lines highlight selected distinctive portfolios. The upward direction on each axis in (**a**) is desirable and a straight line across the top would present an ideal plan; diagonal lines between axes indicate trade-offs, whereas horizontal lines represent synergies. The circle and bar colours in (**b**) and (**c**, **d**) correspond to the lines with the same colours in (**a**). Mt stands for million tonnes, kt stands for kilotonnes, US$mln stands for US dollar millions, vRES stands for variable renewable energy sources, GW stands for giga-watts. Source data are provided as a Source Data file via Zenodo at https://doi.org/10.5281/zenodo.14851474 (ref. 81).

shift to a low-carbon future (the "low emissions" and "low curtailment" portfolios - Figs. 3 and 4). Figure 4 presents the infrastructure expansion by region of the selected portfolios from Fig. 3a. This figure shows how more informed infrastructure planning reduces inequities in electricity access (i.e., adding new generation and transmission capacity and changing electricity allocation priorities in regions with high existing disparities - the Northeast and Northwest regions) while maintaining the same level of emissions and investment cost (e.g., the

"low energy Gini and low emissions" and "low irrigation water Gini and high yield" portfolios - see Figs. 3a and 4). Results also show more infrastructure expansion in the Northwest and Northeast regions across the selected portfolios, with higher intra-regional transmission line expansion than cross-regional transmission line expansion (Fig. 4 and Supplementary Fig. 2). Increased investments in renewables and transmission lines, hydropower reservoir re-operation, and better spatial planning could increase equity in the regional distribution of

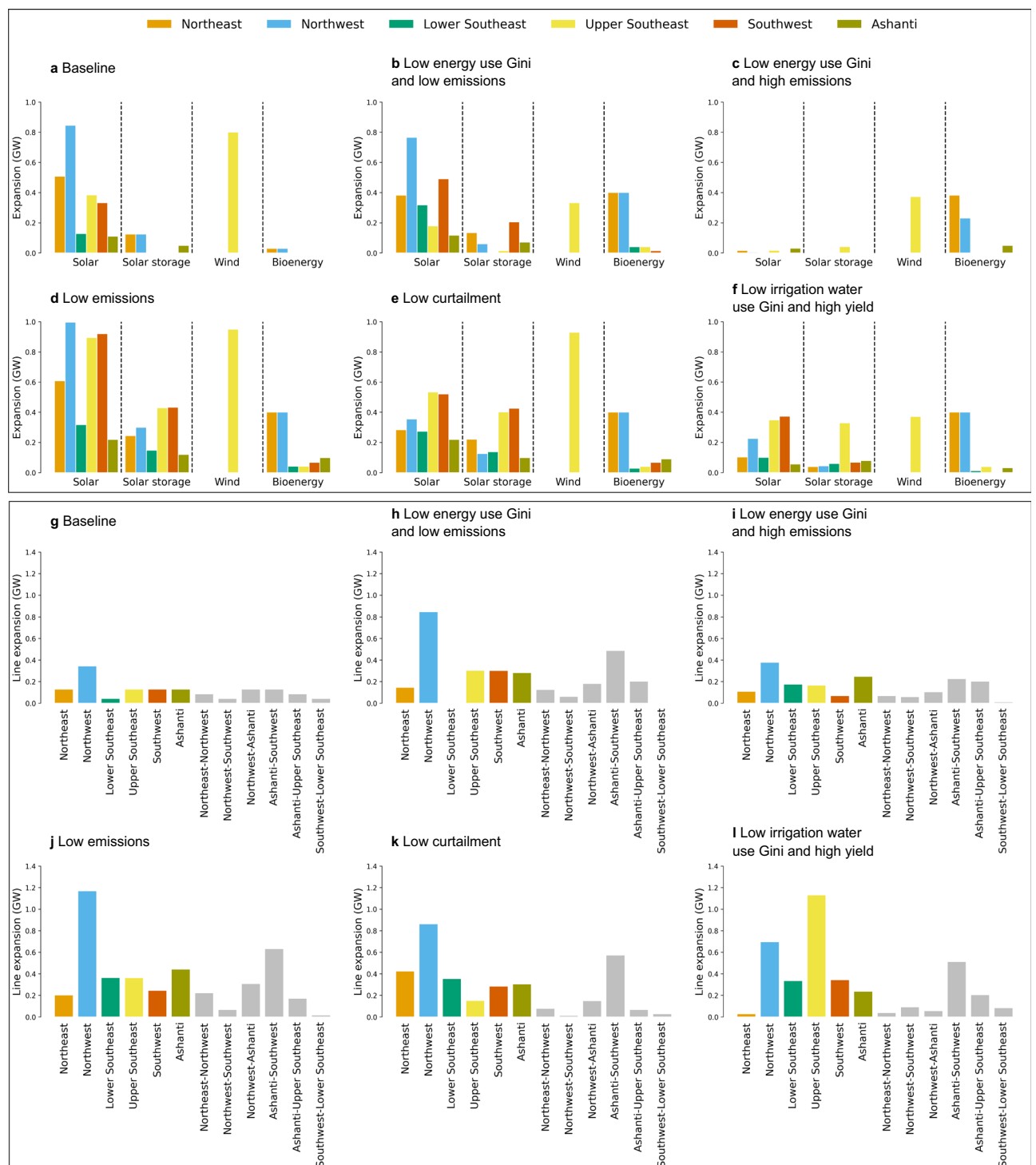

**Fig. 4 | Infrastructure expansion distribution by region.** The top set of figures (**a–f**) show the regional distribution of renewables, and the bottom set of figures (**g–l**) show the regional distribution of transmission line expansion of portfolios selected in the Pareto-efficient portfolios (highlighted coloured lines of Fig. 3a). Solar, wind, solar storage, and bioenergy expansion are higher and more diverse in the "low emissions", "low energy Gini and low emissions", and "low curtailment" portfolios, and lower in the "low energy Gini and high emissions" and "low irrigation water Gini and high yield" portfolios. The figure also shows that there is more infrastructure expansion in the Northwest and Northeast regions across the selected portfolios and that the intra-regional transmission line expansion (sum of first six coloured bars in (**g–l**)) is higher than cross-regional transmission line expansion (sum of last six coloured bars in (**g–l**)), see Supplementary Fig. 2 and Fig. 3b for the aggregated portfolios of infrastructure expansion. GW stands for gigawatts. Source data are provided as a Source Data file via Zenodo at https://doi.org/10.5281/zenodo.14851474 (ref. 81).

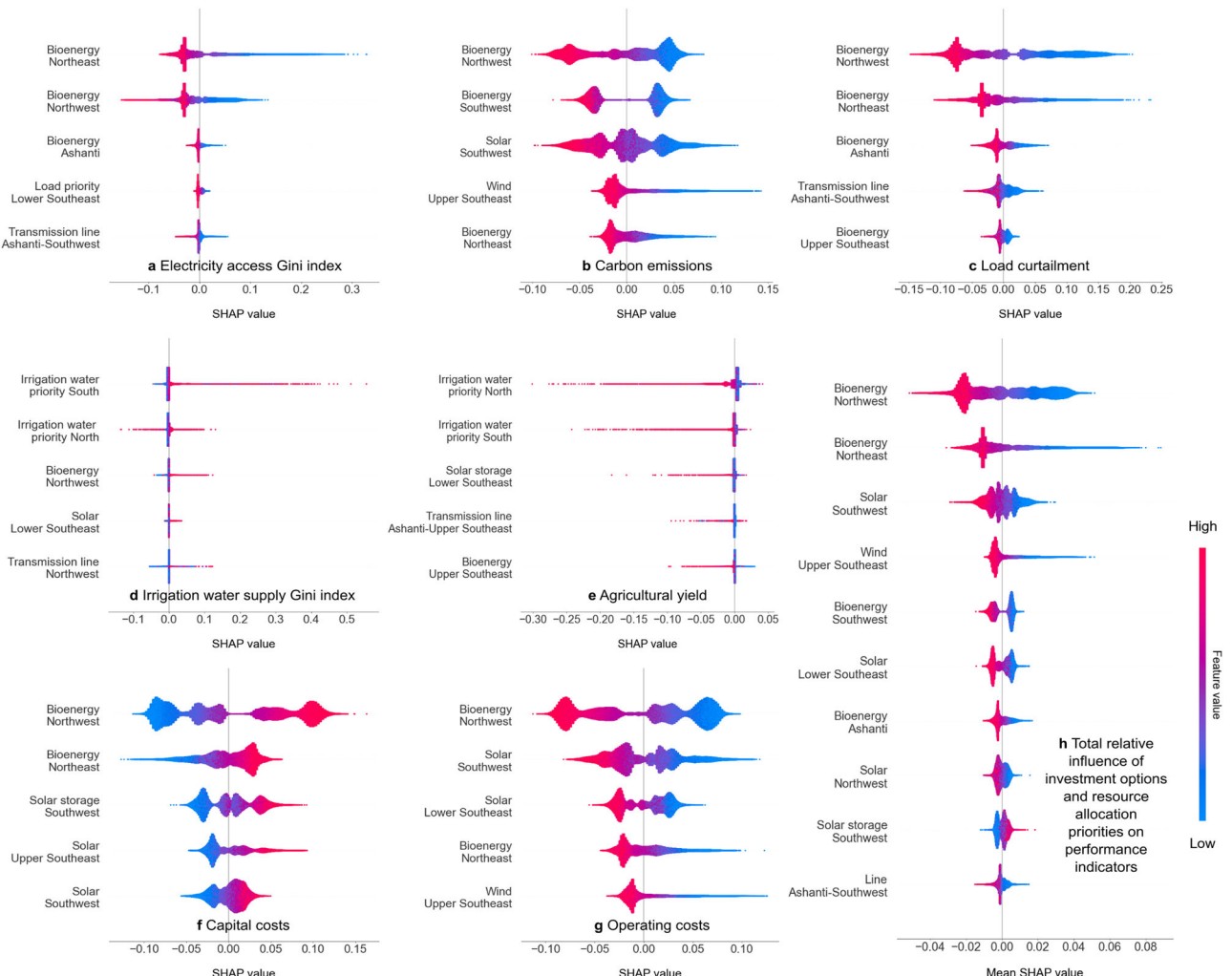

**Fig. 5 | Rankings of investment and resource allocation priority options based on their relative influence on low-carbon energy transition goals. a–g** The five most influential intervention decisions for equity and other water-energy system performance indicators. The x-axes show Shapley values; the y-axes show decision variables. Colour coding indicates the low (blue) to high (red) variable values. The thickness along each decision variable row represents data point density. Shapley values are based on the Shapley additive explanations (SHAP) of machine learning models. The positive and negative Shapley values refer to positive and negative correlations between intervention decisions and energy transition goals. **h** The total relative influence of the top ten intervention options on river basin-power system performance. The total relative influence value in (**h**) is the mean of the Shapley values for the performance indicators in (**a–g**). Source data are provided as a Source Data file via Zenodo at https://doi.org/10.5281/zenodo.14851474 (ref. 81).

(water and energy) services and reduce carbon emissions while improving agricultural performance and meeting future energy service targets (the "low emissions" and "low curtailment" portfolios - Figs. 3 and 4). With low renewable investments, equity can also be improved, but at the cost of higher emissions and load curtailments (the "low energy Gini" portfolios - Figs. 3 and 4).

### Influence of intervention decisions on equity and water-energy system performance

The investment and resource allocation priority options show varying degrees of influence on performance indicators (Fig. 5). Some intervention options are effective at bolstering specific performance indicators, whereas others impact multiple indicators (Fig. 5a–g). The most effective option for equity in electricity access and load curtailment is investing in bioenergy plants in the Northeast and Northwest regions of Ghana (Fig. 5a, c). In contrast, equity in irrigation water supply and agricultural yield are influenced the most by irrigation water allocation priorities (Fig. 5d, e). Results show the most effective choices for improving multiple performance indicators are changing the capacities of bioenergy generation plants in the Northwest and Northeast

regions and investing in the Southwest's solar power and bioenergy and Southeast's wind power (Fig. 5h).

## Discussion

Considering regional benefit distribution equity in multisector infrastructure planning can help guide a country toward a more equitable transition from fossil fuels to lower-emission energy sources while reducing inter-sectoral conflicts across multi-resource systems. A more equitable energy transition will accelerate transformational SDG co-benefits[20] and help reach the Paris Agreement[7]. This paper presents a WEFE artificial intelligence-assisted analytical framework that identifies trade-offs and synergies of multi-dimensionally efficient and more equitable low-carbon water-energy infrastructure and operation portfolios. The framework can help stakeholders identify infrastructure investments and resource allocation policies and evaluate the importance ranking and positive and negative influence of different interventions in achieving energy transition policy goals. Through iterative stakeholder deliberation processes, new or modified intervention options can be set and used to revise intervention portfolios to ensure they appropriately reflect stakeholder aspirations.

Our approach aims to lower national emissions in Ghana through a balanced energy infrastructure system expansion programme (bioenergy, solar, wind, solar storage, and transmission lines), and reservoir re-operation (of Akosombo, Bui, and Pwalugu dams). Ghana's resulting efficient and equitable low-carbon intervention strategies show how considering equity in energy transitions allows reducing both regional service disparities and energy-related greenhouse gas emissions. We show how the spatial distribution of investments drives electricity access equity and that increased investments in renewables and transmission lines, hydropower re-operation, and better spatial planning enhance equity and reduce carbon emissions while improving food production and meeting future energy service goals. With low renewable investments, equity can also be improved, but at the cost of higher emissions and load curtailments. Ghana's power system continues to partially rely on thermal power plants to meet the growing electricity demand despite the expansion of renewable technologies, resulting in relatively high thermal-based carbon emissions. However, adding bioenergy generation (and transmission lines) helps displace thermal power and decreases emissions while increasing equity in service provision in the Ghanaian case. Bioenergy, which uses agricultural crop and forestry residues and municipal waste as an input[50], is recognised as Ghana's leading spatially distributed and dispatchable renewable generation technology. Crop residues alone in Ghana have an estimated bioenergy potential of 75 terajoules (TJ)[50,56]. Other studies have also highlighted the potential of crop residues for bioenergy in Ghana[57–59]. However, increasing bioenergy could compete with food production for land resources if only relying on energy crops for bioenergy[60,61]. Bioenergy production, therefore, could be regulated to prevent a negative impact on food security.

The proposed WEFE infrastructure design approach help analysts and policymakers plan for equitable low-carbon futures. The proposed approach considers the spatial equity of water and energy services alongside emission reduction and other aggregated national infrastructure planning goals to identify efficient, equitable, climate-compatible intervention portfolios for multisector WEFE systems. Pareto-optimal portfolios (Figs. 3 and 4 for Ghana) help identify infrastructure interventions that reduce service inequities and emissions and improve sectoral complementarities. Stakeholders can deliberate the pros and cons of a large set of efficient intervention options to help find a compromise portfolio that manages trade-offs and leverages synergies across multiple competing interests. The machine learning technique helps stakeholders understand how different interventions drive equity, emission, and other system performance targets, as shown for Ghana (Fig. 5). This allows choosing portfolios or revising intervention choices and repeating the process if no consensus can be attained. The proposed approach could be adapted for other countries planning an equitable transition to lower-emission energy sources, especially those with service inequities and interdependent WEFE challenges.

This study did not consider the potential effects of climate change and uncertainties with future projections in the equity-informed planning of WEFE systems. Climate change could impact energy generation, the demand for water and energy, and water resources management. These changes and future uncertainties could have repercussions on the equity of providing water and energy services and inter-sectoral conflicts among multiple resource systems. Addressing these limitations in future research would help identify robust infrastructure strategies for adapting to climate change.

In this paper, we argue for considering equity in shaping future multisector resource systems to reduce service inequities and emissions and better balance regional sectoral benefits. The approach can support countries in phasing out fossil-fuel electricity sources and the transition to carbon neutrality fairly and transparently. As such, it is a relevant approach for developing countries seeking to transition towards lower carbon emissions while at the same time improving

services. Realising an equitable low-carbon future requires considering multiple sectors, their interactions, and the equity implications of adopting different investment and policy intervention packages. Adding equity considerations alongside goals like climate mitigation could help achieve the 2030 agenda for sustainable development and the Paris Agreement.

## Methods

### River system simulator

The river system is modelled using Python Water Resources (Pywr)[45]. Pywr is an open-source Python repository that simulates resource system networks. It allows representing water resources infrastructure using a network structure driven by water supplies and demands and system operating rules. Pywr uses a linear programme to solve water allocations at every simulation time step (weekly in this study) by minimising allocation penalties subject to constraints imposed by operating rules. The Ghana river system model includes existing and planned infrastructure in Ghana. More details on the model can be obtained in the previous publications on the model[43,46].

### Power system simulator

The power system model uses a direct current optimal power flow linear programme[47] to minimise power system costs at each simulation time step (i.e., hourly) subject to equality constraints (power balance at each node) and inequality constraints (power generation and line limits). The power system model is an interconnected network of different energy sources, transmission lines, substations, distribution networks, and electricity demand locations. Energy generators can be represented as nodes in the network, each with their defined constraints, such as maximum power capacity and associated costs. The Ghana power system model simulates the existing and planned hydro, solar, combined solar and storage, wind, bioenergy, and thermal generators[50]. Electricity demand projections, generating resources, capital and operating costs of power plants and transmission lines, and hourly profiles of renewables were taken from Ghana's Power System Master Plan[50]. Hourly electricity demand profiles and transmission data were collected from the Ghanaian national grid operator, the Ghana Grid Company. The long-term annual peak load projections for 2030-2040 from the Ghana Power System Master Plan[50] were used to scale the hourly load profile of 2018 to consider the projected growth in the electricity demand in the country (a twofold increase in the peak demand by 2030 compared to 2018). Capital costs consist of engineering, procurement, construction, start-up, and owner costs (for items like land, cooling infrastructure, administration and associated building, site works, project management, and licenses)[50]. The operating costs of variable renewables (solar and wind) and combined solar and storage are lower than the costs of hydropower, bioenergy, and thermal power plants[50]. The operating costs of thermal generators also differ depending on the technology and fuel types. Hourly power system simulations help capture the variability of intermittent renewables (solar and wind) and energy demand, as well as improve the use of hydropower based on the changing availability of renewables and other energy sources. The power system model dispatches the system's generators based on cost-merit prioritisation where solar, wind, combined solar and storage generators are dispatched first, then hydropower, bioenergy, and lastly, thermal generators. This balances energy production to align with variable energy demand and shares of different energy technologies.

### Integrated river basin-power system simulator

The river and power systems simulators are linked to represent multisector nexus dynamics[43]. The integration of the simulators uses a multi-actor object-oriented Python Network Simulation framework (Pynsim)[48]. The simulators run sequentially with feedback across the simulators (at hydropower generation nodes). The river system

simulator runs its first simulation time step and generates the weekly average hydropower, which is then passed to the power system simulator by Pynsim as a constraint on the maximum power generation capacity for that week. The power system simulator runs for the same week at an hourly simulation time step, constrained by the information provided by the river system simulator. This process is repeated time step by time step until the simulation is completed. The integrated simulation model runs for a period of 10 years, from 2030 to 2040.

### Simulation linked with artificial intelligence design tools

The multisector simulator is connected to an artificial intelligence-assisted multi-objective search algorithm (MOEA) to search for efficient infrastructure intervention portfolios[43]. MOEAs seek the approximately Pareto-optimal solutions to multi-objective optimisation problems by mimicking the natural biological evolution process[62,63]. Supplementary Fig. 3 shows the interaction between the integrated river and power system simulators and the MOEA. The MOEA generates an infrastructure intervention portfolio (i.e., a set of decision variables) and passes it to the integrated water-energy simulator. The integrated simulation model then performs a multi-year simulation to evaluate performance indicators (e.g., regarding energy transition goals) that are used as objectives to be minimised or maximised in the search process. The portfolio and values of the objectives are then stored before the next iteration, when the algorithm generates a new set of variables (a new portfolio) for the simulation model. The search algorithm learns at each iteration from the multi-objective performance of the previous iteration. The iteration between the search algorithm and the integrated simulation model continues until a stopping criterion is met. We allow up to 1.5 million iterations in this design process. Once the stopping criterion is met, a non-dominated sorting process is performed to find a set of best-performing (i.e., Pareto-optimal) intervention portfolios. Pareto-optimal solutions are efficient because they provide the best-achievable compromise among multiple performance objectives that coexist in multi-objective problems. These solutions are non-dominated, indicating no other solutions can outperform or equal them in all objectives at the same time. Stakeholders can consider these optimised solutions to find a compromise that efficiently balances trade-offs among multiple competing interests and leverages synergies. After the search, we used a machine learning method (details provided below) to assess the effectiveness of interventions in achieving performance objectives. Machine learning and interactive data visualisation tools help stakeholders understand, evaluate, and choose a preferred portfolio or iteratively revise the intervention options and repeat the process to obtain alternative, potentially better, intervention portfolios.

We used the Borg MOEA[62,63] to perform the WEFE multi-objective optimisation. Borg can tackle complex nonlinear and nonconcave optimisation problems[63,64]. It integrates various elements from different MOEAs and introduces several new features. It uses ε-dominance archives and ε-progress to maintain a diverse set of Pareto-optimal solutions and monitor convergence speed[62,63]. It employs randomised restarts when convergence slows to revive search through self-adaptive population sizing (of solutions) and dynamically adapts the use of different recombination operators[62,63] to generate new candidate solutions. The evolution process in Borg involves evolving a population of candidate solutions over generations of iterative selection, crossover, and mutation operations. The selection step uses the "survival of the fittest" principle and chooses the best population of solutions based on their performance across the optimisation objectives. The crossover step then combines the characteristics of the selected solutions ("parents") to produce new solutions ("children") by imitating the natural reproduction process. Random mutations are then added to the children to improve variability and adaptability in the new population of solutions. These newly candidate individuals are

then evaluated, and the process continues iteratively, as explained above and in Supplementary Fig. 3.

The convergence of the multi-objective search to the approximately Pareto-optimal solutions was determined by tracking the evolution of the hypervolume[65] for each of four random seeds. Each random seed uses a different starting point for the MOEA search process. The hypervolume measures the volume of the objective space above the non-dominated approximation set. Supplementary Fig. 4 shows that the hypervolume stabilises before the stopping criterion of 1.5 million iterations for all four seeds, thereby indicating convergence.

The design formulation for Ghana includes eight objectives: minimising the electricity access Gini index (Eq. (1)), minimising the carbon emissions from generation (Eq. (2)), minimising the power system load curtailment (Eq. (3)), minimising the irrigation water supply Gini index (Eq. (4)), maximising the agriculture yields (Eqs. (5–8)), minimising the power system capital costs (Eq. (9)), minimising the power system operating costs (Eq. (10)), and maximising the flood recession agriculture benefits (Eqs. (11–13)). The multi-objective search optimises 170 interventions (decisions variables), including the vector of the power system infrastructure expansion, including solar, wind, combined solar and storage, and bioenergy (biomass and biogas) generators, and transmission lines, the vector of operating rules parameters (Eq. (14)) for Akosombo, Bui, and Pwalugu reservoirs, and the vector of the electricity and irrigation water allocation priority parameters.

$$\text{Gini}_{\text{electricity}} = \frac{1}{T} \sum_t \frac{\sum_{i \in \text{RP}} \sum_{j \in \text{RP}} \left| \text{LS/cap}_{i,t} - \text{LS/cap}_{j,t} \right|}{2 \times n^2 \times \overline{\text{LS/cap}}} \quad (1)$$

where, $\text{Gini}_{\text{electricity}}$ is the mean electricity access Gini index of a country, RP is a set of regions of equal population in a country, $\text{LS/cap}_i$ is the load supply per capita of region $i$, $\text{LS/cap}_j$ is the load supply per capita of region $j$, $\overline{\text{LS/cap}}$ is the mean load supply per capita of regions of equal population in a country, $n$ is the number of regions of equal population in a country, $t$ is the simulation time step, and $T$ is the number of simulation years. We calculate the average index value by dividing the sum of the values aggregated in every simulation year and the number of simulation years.

$$\text{Emissions} = \frac{1}{T} \sum_t P_{t,i} \times \sum_i \text{ec}_i^{\text{co}_2} \quad (2)$$

where, $P_{t,i}$ is the generation from the power system model of the generator plant $i$, $\text{ec}_i^{\text{co}_2}$ is the $CO_2$ emission coefficient of the generator plant $i$, $t$ is the simulation time step, and $T$ is the number of simulation years.

$$\text{LC} = \frac{1}{T} \sum_t \sum_n \text{LC}_{t,n} \quad (3)$$

where, LC is the average power system load curtailment, $\text{LC}_{t,n}$ is the load curtailment calculated at every simulation time step $t$ after performing the balance at each bus $n$, and $T$ is the number of simulation years.

$$\text{Gini}_{\text{iws}} = \frac{1}{T} \sum_t \frac{\sum_{i \in \text{RP}} \sum_{j \in \text{RP}} \left| \text{CR}_{i,t} - \text{CR}_{j,t} \right|}{2 \times n^2 \times \overline{\text{CR}}} \quad (4)$$

where, $\text{Gini}_{\text{iws}}$ is the mean irrigation water supply Gini index of a country, RP is a set of regions of equal population in a country, $\text{CR}_i$ is the irrigation water supply curtailment ratio of region $i$, $\text{CR}_j$ is the irrigation water supply curtailment ratio of region $j$, $\overline{\text{CR}}$ is the mean irrigation water supply curtailment ratio of regions of equal population

in a country, $n$ is the number of regions of equal population in a country, $t$ is the simulation time step, and $T$ is the number of simulation years. We calculate the average index value by dividing the sum of the values aggregated in every simulation year and the number of simulation years.

$$Y = \frac{1}{T} \sum_t \sum_n CR_{t,n} \times (A_n \times y_n) \tag{5}$$

$$CR_{t,n} = r_{t,n}/iwr_{t,n} \tag{6}$$

$$iwr_{t,n} = \sum_{ct \in n} cwr_{t,(ct \in n)}/(\alpha_{ct} \times \beta_{ct}) \tag{7}$$

$$cwr_{t,(ct \in n)} = \max\left(0, \left(Kc_{t,(ct \in n)} \times ETo_{t,(ct \in n)} - R_{t,n}\right) \times A_{(ct \in n)}\right) \tag{8}$$

where, $Y$ is the total irrigation yield in tonnes per year estimated using the Food and Agriculture Organization (FAO) Crop Water Requirements method[66], $CR_{t,n}$ is the irrigation water supply curtailment ratio, $A_n$ is the area in hectare (ha) per irrigation scheme $n$, $y_n$ is the annual yield in tonnes per ha per irrigation scheme $n$, $t$ is the simulation time step, $T$ is the number of simulation years, $r_{t,n}$ is the crop water allocated by the model, $iwr_{t,n}$ is the irrigation water requirement for irrigation scheme $n$, $cwr_{t,n}$ is the crop water requirement per irrigation scheme $n$, $\alpha_{ct}$ is the application efficiency (assumed 80%), $\beta_{ct}$ is conveyance efficiency (assumed 70%), and $Kc_{t,(ct \in n)}$, $ETo_{t,(ct \in n)}$, and $R_{t,n}$ are crop water coefficients, reference evapotranspiration in millimetre (mm) per day, and effective rainfall in mm per day obtained from ref. [67].

$R$, $Kc$, and $ETo$ are used in the FAO method[66] to calculate the water requirement for each crop. $ETo$ is the evapotranspiration of a standardised reference crop, while $Kc$ is the factor to change $ETo$ for specific crop water requirements. $R$ is compared with the amount of water a crop needs ($Kc \times ETo$) to determine whether irrigation is required for that crop. Crops that are considered vary for each irrigation scheme and include rice, maize, sugar cane, beans, tomatoes, and fresh vegetables. The total irrigation water requirement for each irrigation scheme is calculated by assuming the overall irrigation efficiency for surface irrigation. The level of irrigation water supply compared to the water required is then used to estimate the total irrigation yield.

$$CAPEX = \sum_i cc_i \times cap_i \tag{9}$$

where, $cc_i$ is the technology capital cost, and $cap_i$ is the new infrastructure capacity.

$$OPEX = \frac{1}{T} \sum_t \sum_i oc_i \times P_{t,i} \tag{10}$$

where, $oc_i$ is the operating cost of the generator technology $i$, $P_{t,i}$ is the power output from the generator technology $i$, $t$ is the simulation time step, and $T$ is the number of simulation years.

$$FRA = \sum_n \beta_{FRA} \times Y_n \tag{11}$$

$$Y_n = A_n^f q_n^{FRA} f_{FRA} C_y \tag{12}$$

$$q_n^{FRA} = \text{mean}[\max(q_{t,n}^{Aug}, q_{t,n}^{Sep})] \tag{13}$$

where, FRA is the flood recession agriculture benefits in US$ depending on the flooding of the floodplain during the rainy peak period (July to September for northern Ghana), $\beta_{FRA}$ is the average crop market price at US$ 1222 per tonne[68], $Y_n$ is the total FRA yield in tonnes per year, $A_n^f$ is the flooded area in ha, $q_n^{FRA}$ is the average flow in August or September in the model simulation time, $f_{FRA}$ is the suitability factor[69], $C_y$ is the crop yield in tonnes per ha assuming a typical crop mix of beans, maize, soya, bambara beans, millet, and groundnuts[70], and $q_{t,n}$ is the average flow in August and September.

We used Gaussian radial basis functions (RBFs) to model reservoir operating rules. The use of RBFs in representing infrastructure operating rules, including reservoir storage and release decisions, have shown good performance[71–73]. The Gaussian RBF ($\varphi$) is defined by Eq. (14).

$$\varphi(x) = \sum_{i=1}^n w_i \times \exp\left[-\sum_{j=1}^m \frac{\left(x_j - c_{i,j}\right)^2}{b^2_{i,j}}\right] \tag{14}$$

where, $m$ is the number of input variables $x$ (reservoir storage and time), $n$ is the number of RBFs (four), $w_i$ is the weight of the $i$th RBF, and $c_{i,j}$ and $b_{i,j}$ are the $m$-dimensional centres and radius vectors of the $i$th RBF, respectively. The centres and radius take values in $c_{i,j} \epsilon [-1,1]$ and $b_{i,j} \epsilon [0, 1]$, and $w_i \epsilon [0, 1]$ with $\sum_{i=1}^n w_i = 1$. More details can be found in the ref. [71]. The time of year and storage in the reservoir are mapped to the release rule at each time step of the simulation period. This allows to dynamic adaptation of releases based on changing reservoir volumes. The multi-objective search process (MOEA) optimises the monthly storage volumes and releases (24 decision variables) for each reservoir to determine the most efficient (Pareto-optimal) operating rules. The MOEA iteratively optimises the release rules alongside infrastructure expansion, such as solar, wind, combined solar and storage, bioenergy, and transmission lines, and energy and water allocation priorities. Hydropower reservoirs can provide flexibility and stability for power grids, and their improved operations can support the transition to clean energy[43,74–76]. The MOEA explores a range of different infrastructure interventions while also adjusting reservoir release decisions to find the best-achievable compromise among equity, emissions, and other water-energy system performance objectives. This way MOEA solutions adapt to the changing shares of different energy technologies and allocate resources to improve overall system performance (see Supplementary Fig. 3 for a description of the interaction between the integrated system model and the MOEA search).

We used the Random Forest Regression machine learning model[77] from the Scikit-learn Python library[78] and the SHAP[49,79,80] to determine the influence of intervention decisions on equity and other water-energy system performance indicators. The MOEA typically yields a large set (up to thousands) of optimal solutions for large problems like the Ghana case study. While common patterns can potentially be identified from directly exploring Pareto-optimal solutions or via cluster analysis, the SHAP analysis of tree-based machine learning models[49,79] helps explain the relationships between intervention decisions and the MOEA performance measures. It provides the importance ranking of different interventions and also the direction and magnitude of their influence in achieving performance objectives. The outcomes of the MOEA are used to train a Random Forest model for each performance objective. The features of each Random Forest model are the decision variables representing different investment and resource allocation priority options, and the target is each of the optimisation objectives. We divide the data into the training set (80% of the data) and the testing set (20% of the data). The training dataset is used to train 100 tree estimators, and the testing dataset is used to test the performance of the Random Forest models. We also test the performance of each model for different maximum tree depths ranging

from 1 to 70 to select the best-fit tree depth for each model to avoid overfitting or underfitting models. Supplementary Fig. 5 depicts the performance of the machine learning models for the training and testing sets with different maximum tree depths and the selected maximum tree depth for each model.

## Data availability
The data for the river system model can be made available upon presentation of the necessary permission from the Ghana Council for Scientific and Industrial Research - Water Research Institute that owns the data. The data for the power system model are free to access and can be found through the link: https://energycom.gov.gh/planning/ipsmp/ipsmp-2018/gh-ipm-v1-2018-assumptions-model-results. Source data for the figures are available via Zenodo at https://doi.org/10.5281/zenodo.14851474 (ref. 81).

## Code availability
The Pywr library used to develop the river system model is open-source and freely available at: https://github.com/pywr/pywr. The Pyenr library used to develop the power system model is open-source and freely available at: https://github.com/pywr/pyenr. The Pynsim library is open-source and freely available at: https://github.com/UMWRG/pynsim. The Random Forest Regression model is open-source and freely available at: https://github.com/scikit-learn/scikit-learn. The SHAP python package is open-source and freely available at: https://github.com/shap/shap. The FastTreeSHAP python package is a fast implementation of SHAP for tree-based models. The FastTreeSHAP python package is open-source and freely available at: https://github.com/linkedin/FastTreeSHAP.

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

## Acknowledgements

The authors acknowledge UKRI Global Challenge Research Fund funding through the "Future Design and Assessment of water–energy–food–environment Mega Systems" research project (ES/P011373/1), which was applied for and administered by J.J.H. The

University of Manchester financially supported this work: A.A. received a Dean's Doctoral Scholarship from the University of Manchester. The authors acknowledge the use of the Computational Shared Facility (CSF) and High-Performance Computing (HPC) of the University of Manchester and associated support services.

## Author contributions

A.A. wrote the original manuscript; A.A. led visualisation of the results; A.A., M.B., J.J.H., J.M.G., M.P., E.A.M.C., A.B-B., J.A., M.E., and E.O. reviewed and edited the manuscript; A.A. performed simulation and optimisation works; J.M.G. and A.A. developed Ghana's water-energy model; A.A. and J.M.G. linked the water-energy model with artificial intelligence algorithms; A.A., M.B., J.J.H., J.M.G., and M.E. conceptualised the study; J.J.H. and M.B. supervised the work; A.A., M.B., J.J.H., J.M.G., M.P., E.A.M.C., A.B-B., J.A., M.E., and E.O. contributed to validation and interpretation of the results.

## Competing interests

The authors declare no competing interests.
