## [Transparent Peer Review file · Nature Communications]

Delivering equity in low-carbon multisector infrastructure planning

Corresponding Author: Professor Julien Harou

Version 0:

Reviewer comments:

Reviewer #1

See the attachment

(Remarks to the Author)

Reviewer #2

(Remarks to the Author)

1. Line 167, what are the “system operating rules” reservoirs or hydropower stations? How to determine the optimal operating rules? Operating rules are normally used for mid- and long-term operations (more than one day), how to consider the intraday operation process of reservoirs? Hydropower is a key part of the energy system for its flexibility. According to previous related studies [1-3], the hydropower operation is important and adjustable in system simulation, which may significantly alter the planning. Are some kinds of dynamic optimal operation strategies for hydropower stations that can adapt to the changing shares of other energies being considered?

References:

- [1] Liu, Z., He, X. Balancing-oriented hydropower operation makes the clean energy transition more affordable and simultaneously boosts water security. *Nat Water* 1, 778–789 (2023). <https://doi.org/10.1038/s44221-023-00126-0>
- [2] Sterl, S., Fadly, D., Liersch, S. et al. Linking solar and wind power in eastern Africa with operation of the Grand Ethiopian Renaissance Dam. *Nat Energy* 6, 407–418 (2021). <https://doi.org/10.1038/s41560-021-00799-5>
- [3] Conway, D., Dalin, C., Landman, W.A. et al. Hydropower plans in eastern and southern Africa increase risk of concurrent climate-related electricity supply disruption. *Nat Energy* 2, 946–953 (2017). <https://doi.org/10.1038/s41560-017-0037-4>

2. What is the applicability of the proposed framework? Can this method be applied in other countries? For a more complex case study, how to consider other types of energies (e.g., nuclear, offshore wind, hydrogen)? Line 223, what is the total number of decision variables? For a case study with more hydropower stations and more complex topological relationships, how to deal with the high optimization dimensionality?

3. What are the innovations in findings and methods of this study compared to the authors' previous study? (Designing diversified renewable energy systems to balance multisector performance).

4. How to select the equitable transition plans? Does this question need methods of game theory to achieve equity?

5. How the study considers the energy prices under different portfolios?

6. Why use the Random Forest Regression and SHAP method? Is it possible to directly analyze the influence of intervention decisions through the optimization model?

7. Line 156-160, this sentence is too long.

Reviewer #3

(Remarks to the Author)

The manuscript presents a compelling argument for considering spatial equity in water and energy service provision while planning low-carbon energy transitions, especially in developing countries with multi-sector interdependencies and regional disparities. The proposed analytical framework combining integrated river basin-power system simulation with artificial intelligence design tools is innovative and well-described. The case study of Ghana demonstrates the utility of the framework and the trade-offs involved in achieving equitable low-carbon futures.

The manuscript addresses an important issue in sustainable development. I do have one major comment that requires some additional clarifications, expansions, and refinements, and a couple of minor comments. However, overall, I believe the paper could make a valuable contribution to the literature on equitable low-carbon energy transitions and multi-sector infrastructure planning. Below are my comments:

Major:

- Food: While the paper outlines objectives related to maximizing agricultural yield and maximizing the flood recession agricultural benefit (Equations 5 and 8), the discussion around the direct implications of the infrastructure changes on food systems remains somewhat underdeveloped. Particularly, the interactions between energy infrastructure, specifically bioenergy, and food systems, need deeper exploration.

In lines 322-323 & 372-374, the manuscript identifies investing in bioenergy plants in the Northeast and Northwest regions of Ghana as an effective strategy for improving equity in electricity access. However, it's crucial to consider the broader implications of such investments. Since agriculture is mentioned as an input, research indicates that bioenergy production can compete with food production for land resources, potentially increasing food prices if food crops are used as inputs for bioenergy (see <https://link.springer.com/article/10.1007/s10584-020-02838-8> and <https://www.sciencedirect.com/science/article/abs/pii/S0360544210002896?via=ihub>)

This competition can exacerbate food security challenges, particularly in regions already vulnerable to food access inequalities. The authors could consider expanding the discussion on potential trade-offs between increased bioenergy production and food availability, an essential component of the water-energy-food nexus and equity implications.

Minor:

1. Line 81. The International Renewable Energy Agency (IRENA) abbreviation is incorrect: "IREA"
2. Figures 2, 3, and 4, the captions are in bold, which is effective and clear. However, the descriptions are formatted in normal text that matches the font of the body text. This can lead to confusion, making it difficult for readers to discern where the figure description ends and the main body text resumes.
3. Consider expanding the discussion section to further elaborate on the implications of their findings for policy-making and potential limitations. For example, the authors could elaborate on the implications of the findings for policymakers and practitioners, i.e., a detailed discussion of how this research could inform infrastructure planning, policy-making, and sustainable development goals in Ghana and how it could be adapted for other regions or different scales of analysis. Potential limitations, i.e., Uncertainties associated with future projections, climate change impacts, etc.
4. Equation 5 could benefit from additional explanations or references to clarify its components and the rationale behind their formulation.

Version 1:

Reviewer comments:

Reviewer #1

(Remarks to the Author)

The authors of this article have made good revisions or supplemented explanations one by one based on the comments. I recommend publishing this article if the code is feasible.

(Remarks on code availability)

Unable to access this page (<https://github.com/pywr/pywr>)

Reviewer #3

(Remarks to the Author)

I have no more comments. All my comments have been addressed to my satisfaction.

Congratulations to the authors.

(Remarks on code availability)

There are two distinct codebases in this project: one for the simulation and another for postprocessing. While I have not personally installed or tested either codebase, both are open-source. The postprocessing codes utilize general purpose, open-source libraries that are freely available to everyone.

Response to the reviewers' comments

We appreciate the reviewers' feedback and suggestions. Below are our point-by-point responses (blue text) to their comments (black text). The blue text highlighted in yellow indicates the text included in the revised manuscript, and the line numbers correspond to the revised version of the manuscript without tracked changes.

Reviewer 1

The consideration of low-carbon and equity multisector infrastructure planning is a hot and difficult issue in the world. This study proposes an analytical framework, taking Ghana as the case study area, considering the regional distribution of basic services such as water and energy, and combining comprehensive simulation of watershed power systems with artificial intelligence design tools. By determining effective infrastructure intervention combinations and their implicit trade-offs between spatial equity, carbon emissions, food production, and river ecosystem performance in providing water and energy services, demonstrate the utility of this framework for Ghana. This study has important theoretical and practical significance for achieving carbon neutrality goals in the fields of water resources and energy. I suggest making major revisions before accepting. The following opinions and suggestions are for reference:

(1) The main text only uses 'Ghana' as a case study area for application analysis. It is recommended to add relevant content such as 'Ghana' or 'Ghana as a case study area' to the title.

We thank the reviewer for this comment. We used a general title (without naming the case-study area) to highlight the generality of the proposed design approach. The proposed approach is applicable in a range of countries keen on an equitable transition to lower-emission energy sources, especially those with high existing service inequities and complex multisector interdependencies and challenges. A title including the case-study location could be feasible, but this would in our view reduce the paper's impact because the framework is the innovation here, the case study is there primarily to demonstrate the framework.

(2) In the section on ‘Equity as a policy goal in low-carbon multisector infrastructure planning’, it is pointed out that ‘multisector infrastructure’ is somewhat fuzzy in the main text; Additional clarification is needed on the specific departments involved. If there are many departments, they can be listed in the relevant table as an attachment.

We appreciate the reviewer’s comment and understand that by ‘departments’, they refer to components included in the multisector infrastructure system. The interlinked river basin-power system model is used to represent spatial water-energy-food-ecosystem (WEFE) dynamics. The river system model represents water resources infrastructure (e.g., dams) using a network structure including water supplies (hydrological inflows) and demands (e.g., municipal and irrigation water requirements and hydropower) and system operating rules. The power system model simulates network connectivity, different power generation types including renewable energy generators, and electricity demand profiles. We have extended the text on the interlinked river basin-power system model under the ‘Equity as a policy goal in low-carbon multisector infrastructure planning’ heading to indicate the main components of the multisector WEFE simulation system in lines 174 to 179, as follows:

The integrated WEFE infrastructure system simulator simulates managed rivers (with river inflows, existing and planned water storage reservoirs, hydropower, municipal water supply abstractions, irrigation, and flood recession agriculture) and energy systems (solar power, combined solar and storage, wind power, bioenergy, thermal power generators, electricity transmission lines, and electricity demands).

A table summarising the type and number of nodes used in the Ghana integrated simulation model is now provided in the supplementary material (Supplementary Table S1, see below). This table is referred in lines 213 to 215 in the ‘Equitable low-carbon energy transition in an African case’ section.

Supplementary Table S1: Type and number of nodes in the Ghana multisector infrastructure system simulation model.

Model	Node type	Number of nodes
River system	Catchment inflow	9
	Reservoir	9
	Hydropower*	6
	Municipal water supply	5
	Irrigation	10
	Flood recession farming	1
	Flood recession pond fishing	1
	Environmental flow	1
Power system	Solar	19
	Combined solar and storage	11
	Wind power	2
	Bioenergy	7
	Hydropower	4
	Thermal generators	24
	Substations	23
	Transmission lines	33
	Electricity demand	22
*These are hydropower nodes within the Volta River basin, including those located in Ghana.		

(3) Lines 36-37, in the abstract section, which points out that ‘The utility of the framework is demonstrated for Ghana by identifying the most efficient infrastructure intervention portfolios and’, but this section in the main text is somewhat fuzzy; Further explanation is needed to determine the key process for determining the ‘most effective’ reasoning; Provide sufficient detailed explanations.

We thank the reviewer for this comment. In our response to the next comment below, we have addressed this comment regarding the process of finding the most efficient set of intervention solutions.

(4) Lines 443-444, the article points out that ‘A maximum of 1.5 million iterations is specified as a stopping criterion in the design process’, and several key iteration processes need to be provided, along with clear explanations of the corresponding algorithms; Please provide key details.

We have addressed the above two comments together. We have revised the ‘Simulation-based artificial intelligence-assisted multisector system design and machine learning’ section to provide further explanations on the process of finding the most efficient interventions. We have also provided more detail on the Borg multi-objective evolutionary algorithm (MOEA). We have included a new figure in the

supplementary material (Supplementary Fig. S3, see below) that shows the iteration between the integrated river basin-power system model and the MOEA. The updated text is in lines 492 to 546 and reads:

The multisector simulator is connected to an artificial intelligence-assisted multi-objective search algorithm (MOEA) to search for efficient infrastructure intervention portfolios⁴². MOEAs seek the approximately Pareto-optimal solutions to multi-objective optimisation problems by mimicking the natural biological evolution process^{61,62}. Supplementary Fig. S3 shows the interaction between the integrated river and power system simulators and the MOEA. The MOEA generates an infrastructure intervention portfolio (i.e., a set of decision variables) and passes it to the integrated water-energy simulator. The integrated simulation model then performs a multi-year simulation to evaluate performance indicators (e.g., regarding energy transition goals) that are used as objectives to be minimised or maximised in the search process. The portfolio and values of the objectives are then stored before the next iteration, when the algorithm generates a new set of variables (a new portfolio) for the simulation model. The search algorithm learns at each iteration from the multi-objective performance of the previous iteration. The iteration between the search algorithm and the integrated simulation model continues until a stopping criterion is met. We allow up to 1.5 million iterations in this design process. Once the stopping criterion is met, a non-dominated sorting process is performed to find a set of best-performing (i.e., Pareto-optimal) intervention portfolios. Pareto-optimal solutions are efficient because they provide the best-achievable compromise among multiple performance objectives that coexist in multi-objective problems. These solutions are non-dominated, indicating no other solutions can outperform or equal them in all objectives at the same time. Stakeholders can consider these optimised solutions to find a compromise that efficiently balances trade-offs among multiple competing interests and leverages synergies. After the search, we used a machine learning method (details provided below) to assess the effectiveness of interventions in achieving performance objectives. Machine learning and interactive data visualisation tools help stakeholders understand, evaluate, and choose a

preferred portfolio or iteratively revise the intervention options and repeat the process to obtain alternative, potentially better, intervention portfolios.

We used the Borg MOEA^{61,62} to perform the WEFE multi-objective optimisation. Borg can tackle complex nonlinear and nonconcave optimisation problems^{62,63}. It integrates various elements from different MOEAs and introduces several new features. It uses ϵ -dominance archives and ϵ -progress to maintain a diverse set of Pareto-optimal solutions and monitor convergence speed^{61,62}. It employs randomised restarts when convergence slows to revive search through self-adaptive population sizing (of solutions) and dynamically adapts the use of different recombination operators^{61,62} to generate new candidate solutions. The evolution process in Borg involves evolving a population of candidate solutions over generations of iterative selection, crossover, and mutation operations. The selection step uses the “survival of the fittest” principle and chooses the best population of solutions based on their performance across the optimisation objectives. The crossover step then combines the characteristics of the selected solutions (“parents”) to produce new solutions (“children”) by imitating the natural reproduction process. Random mutations are then added to the children to improve variability and adaptability in the new population of solutions. These newly candidate individuals are then evaluated, and the process continues iteratively, as explained above and in Supplementary Fig. S3.

The convergence of the multi-objective search to the approximately Pareto-optimal solutions was determined by tracking the evolution of the hypervolume⁶⁴ for each of four random seeds. Each random seed uses a different starting point for the MOEA search process. The hypervolume measures the volume of the objective space above the non-dominated approximation set. Supplementary Fig. S4 shows that the hypervolume stabilises before the stopping criterion of 1.5 million iterations for all four seeds, thereby indicating convergence.

Supplementary Fig. S3: Flowchart on the iteration between the integrated river basin-power system model and the multi-objective evolutionary algorithm (MOEA) for identifying non-dominated, approximately Pareto-optimal infrastructure intervention portfolios.

(5) Suggest checking of the figures appearing in the text and supplementary materials, and making modifications to them, such as Fig. S2, the differentiation between the vertical axis curves is not high, and it needs to be redrawn more clearly.

We thank the reviewer for the comment. We have checked all the figures. Supplementary Fig. S4 (i.e., Fig. S2 as per old numbering) has been revised using different coloured lines to ensure each curve is distinguishable. This figure shows the evolution of the hypervolume for four random seeds to confirm the convergence to an approximately Pareto-optimal set of solutions.

Supplementary Fig. S4: Evolution of the hypervolume for four random seeds. Each random seed represents a unique starting point for the search algorithm and is shown by a different coloured line in this figure.

(6) It is recommended to conduct a comprehensive check on the citation of references in the text. For example, the article points out that MOEA uses references [55-58] to simulate the natural biological evolution process, and later uses references [59, 60, 60, 61, 62]. There are too many references used, and the evolutionary algorithm used in this study may only have 1-3 references. Please explain clearly; It is necessary to write how this method was applied to this study, as the text is somewhat vague; Please remove any unused references.

We appreciate the reviewer's comment. We have checked the references. The MOEA section has been amended (lines 492 to 546) and now includes further information on the simulation-based multi-objective search process, including the Borg MOEA, the optimisation algorithm used in our study. We have removed the first four MOEA references. These references pointed to general literature on evolutionary algorithms. Please see the response above addressing comments 3 and 4.

(7) Lines 312-315, in Figure 4, 'The figure also shows that there is more infrastructure expansion in the Northwest and Northeast regions across the selected portfolios and that the intra-regional transmission line expansion (sum of first six coloured bars in a2-f2) is higher than cross-regional transmission line expansion

(sum of last six coloured bars in a2-f2)'. It is difficult to see that the expansion of transmission lines within the region is higher than that of cross regional transmission lines. Please further explain.

We thank the reviewer for the comment. We have now added another figure in the supplementary material, which is referred in the revised manuscript in lines 321 and 322. This new Supplementary Fig. S2 is provided below and presents the aggregated distribution of intra- and cross-regional transmission line expansion of the baseline and selected Pareto-efficient portfolios (highlighted coloured lines of Fig. 3a). The figure shows that the intra-regional transmission line expansion is higher than cross-regional transmission line expansion for the baseline and selected efficient portfolios.

Supplementary Fig. S2. Intra- and cross-regional electrical transmission line expansion for Ghana of the baseline and selected efficient portfolios of Fig. 3a. This figure shows that the intra-regional transmission line expansion is higher than cross-regional transmission line expansion for the baseline and selected Pareto-efficient portfolios. The circle colours correspond to the lines with the same colours in Fig. 3a.

(8) Although this article argues hot research questions, the work involves multiple departments and fields, which is quite complex. It is recommended to further summarize the innovation and contribution of this article in the discussion section.

We thank the reviewer for this comment. We have expanded the discussion section to briefly summarise the innovation and contribution of our work. The new/revised text also provides further insights into the implications and limitations of our study, as suggested by Reviewer 3. This new/revised text is in lines 366 to 416 in the discussion section and reads:

Our approach aims to lower national emissions in Ghana through a balanced energy infrastructure system expansion programme (bioenergy, solar, wind, solar storage, and transmission lines), and reservoir re-operation (of Akosombo, Bui, and Pwalugu dams). Ghana's resulting efficient and equitable low-carbon intervention strategies show how considering equity in energy transitions allows reducing both regional service disparities and energy-related greenhouse gas emissions. We show how the spatial distribution of investments drives electricity access equity and that increased investments in renewables and transmission lines, hydropower re-operation, and better spatial planning enhance equity and reduce carbon emissions while improving food production and meeting future energy service goals. With low renewable investments, equity can also be improved, but at the cost of higher emissions and load curtailments. Ghana's power system continues to partially rely on thermal power plants to meet the growing electricity demand despite the expansion of renewable technologies, resulting in relatively high thermal-based carbon emissions. However, adding bioenergy generation (and transmission lines) helps displace thermal power and decreases emissions while increasing equity in service provision in the Ghanaian case. Bioenergy, which uses agricultural crop and forestry residues and municipal waste as an input⁴⁹, is recognised as Ghana's leading spatially distributed and dispatchable renewable generation technology. Crop residues alone in Ghana have an estimated bioenergy potential of 75 TJ^{49,55}. Other studies have also highlighted the potential of crop residues for bioenergy in Ghana^{56–58}. However, increasing bioenergy could compete with food production for land resources if only relying on energy crops for bioenergy^{59,60}. Bioenergy production, therefore, could be regulated to prevent a negative impact on food security.

The proposed WEFE infrastructure design approach help analysts and policymakers plan for equitable low-carbon futures. The proposed approach considers the spatial equity of water and energy services alongside emission reduction and other aggregated national infrastructure planning goals to identify efficient equitable climate-compatible intervention portfolios for multisector WEFE systems. Pareto-optimal portfolios (Figs. 3 and 4 for Ghana) help identify infrastructure interventions that reduce service inequities and emissions and improve sectoral complementarities. Stakeholders can deliberate the pros and cons of a large set of efficient intervention options to help find a compromise portfolio that manages trade-offs and leverages synergies across multiple competing interests. The machine learning technique helps stakeholders understand how different interventions drive equity, emission, and other system performance targets, as shown for Ghana (Fig. 5). This allows choosing portfolios or revising intervention choices and repeating the process if no consensus can be attained. The proposed approach could be adapted for other countries planning an equitable transition to lower-emission energy sources, especially those with service inequities and interdependent WEFE challenges.

This study did not consider the potential effects of climate change and uncertainties with future projections in the equity-informed planning of WEFE systems. Climate change could impact energy generation, the demand for water and energy, and water resources management. These changes and future uncertainties could have repercussions on the equity of providing water and energy services and inter-sectoral conflicts among multiple resource systems. Addressing these limitations in future research would help identify robust infrastructure strategies for adapting to climate change.

Reviewer 2

1. Line 167, what are the “system operating rules” reservoirs or hydropower stations? How to determine the optimal operating rules? Operating rules are normally used for mid- and long-term operations (more than one day), how to consider the intraday operation process of reservoirs? Hydropower is a key part of the energy system for its flexibility. According to previous related studies [1-3], the hydropower operation is important and adjustable in system simulation, which may significantly alter the planning. Are some kinds of dynamic optimal operation strategies for hydropower stations that can adapt to the changing shares of other energies being considered?

References:

- [1] Liu, Z., He, X. Balancing-oriented hydropower operation makes the clean energy transition more affordable and simultaneously boosts water security. *Nat Water* 1, 778–789 (2023). <https://doi.org/10.1038/s44221-023-00126-0> [doi.org]
- [2] Sterl, S., Fadly, D., Liersch, S. et al. Linking solar and wind power in eastern Africa with operation of the Grand Ethiopian Renaissance Dam. *Nat Energy* 6, 407–418 (2021). <https://doi.org/10.1038/s41560-021-00799-5> [doi.org]
- [3] Conway, D., Dalin, C., Landman, W.A. et al. Hydropower plans in eastern and southern Africa increase risk of concurrent climate-related electricity supply disruption. *Nat Energy* 2, 946–953 (2017). <https://doi.org/10.1038/s41560-017-0037-4> [doi.org]

We thank the reviewer for the comment. We have included an explanation in the methods section regarding reservoir operating rules (i.e., equation 9). We hope this added text helps readers understand how reservoir rules are optimised in our design problem. The new/revised text is in lines 623 to 648 and reads:

We used Gaussian radial basis functions (RBFs) to model reservoir operating rules. The use of RBFs in representing infrastructure operating rules, including reservoir storage and release decisions, have shown good performance^{70–72}. The Gaussian RBF (φ) is defined as follows:

$$\varphi(x) = \sum_{i=1}^n w_i \times \exp\left[-\sum_{j=1}^m \frac{(x_j - c_{i,j})^2}{b_{i,j}^2}\right] \quad (9)$$

where, m is the number of input variables x (reservoir storage and time), n is the number of RBFs (four), w_i is the weight of the i th RBF, and $c_{i,j}$ and $b_{i,j}$ are the m -dimensional centres and radius vectors of the i th RBF, respectively. The centres and radius take values in $c_{i,j} \in [-1,1]$ and $b_{i,j} \in [0,1]$, and $w_i \in [0,1]$ with $\sum_{i=1}^n w_i = 1$. More details can be found in ref.⁷⁰. The time of year and storage in the reservoir are mapped to the release rule at each time step of the simulation period. This allows to dynamically adapt releases based on changing reservoir volumes. The multi-objective search process (MOEA) optimises the monthly storage volumes and releases (24 decision variables) for each reservoir to determine the most efficient (Pareto-optimal) operating rules. The MOEA iteratively optimises the release rules alongside infrastructure expansion, such as, solar, wind, combined solar and storage, bioenergy, and transmission lines, and energy and water allocation priorities. Hydropower reservoirs can provide flexibility and stability for power grids and their improved operations can support the transition to clean energy^{42,73–75}. The MOEA explores a range of different infrastructure interventions while also adjusting reservoir release decisions to find the best-achievable compromise among equity, emissions, and other water-energy system performance objectives. This way MOEA solutions adapt to the changing shares of different energy technologies and allocate resources to improve overall system performance (see Supplementary Fig. S3 for a description of the interaction between the integrated system model and the MOEA search).

Below is the brief description of the integrated simulation model. The text highlighted in yellow indicates the new text (lines 468 to 475 in the revised manuscript).

The integrated simulation model combines the river and power system simulators. The river system simulator runs at a weekly time step for a simulation horizon of 10 years and uses the operating rules of reservoirs mapped via the RBFs to constrain water allocation decisions. The river system simulator first generates the weekly average hydropower, which is then passed to the power system simulator as a

constraint on the maximum power generation capacity for that week. The power system simulator then runs for the same week at an hourly time step, constrained by the information provided by the river system simulator. This process is repeated until the simulation is completed. Hourly power system simulations help capture the variability of intermittent renewables (solar and wind) and energy demand, as well as improve the use of hydropower based on the changing availability of renewables and other energy sources. The power system model dispatches the system's generators based on cost-merit prioritisation where solar, wind, combined solar and storage generators are dispatched first, then hydropower, bioenergy, and lastly, thermal generators. This balances energy production to align with variable energy demand and shares of different energy technologies.

2. What is the applicability of the proposed framework? Can this method be applied in other countries? For a more complex case study, how to consider other types of energies (e.g., nuclear, offshore wind, hydrogen)? Line 223, what is the total number of decision variables? For a case study with more hydropower stations and more complex topological relationships, how to deal with the high optimization dimensionality?

We thank the reviewer for these questions. We have included new text in the revised manuscript to address these points.

Our proposed framework helps analysts and policymakers seek equitable climate-compatible futures. Its adaptability ensures relevance and applicability in a range of countries keen on an equitable transition to lower-emission energy sources, especially those with high existing service inequities and complex multisector interdependencies. We have expanded the discussion section to highlight the applicability of our work. This new and revised text in the discussion section also further specifies the innovation, contribution, implications, and limitations of our study. This new/revised text reads (lines 366 to 416):

Our approach aims to lower national emissions in Ghana through a balanced energy infrastructure system expansion programme (bioenergy, solar, wind, solar storage, and transmission lines), and reservoir re-operation (of

Akosombo, Bui, and Pwalugu dams). Ghana's resulting efficient and equitable low-carbon intervention strategies show how considering equity in energy transitions allows reducing both regional service disparities and energy-related greenhouse gas emissions. We show how the spatial distribution of investments drives electricity access equity and that increased investments in renewables and transmission lines, hydropower re-operation, and better spatial planning enhance equity and reduce carbon emissions while improving food production and meeting future energy service goals. With low renewable investments, equity can also be improved, but at the cost of higher emissions and load curtailments. Ghana's power system continues to partially rely on thermal power plants to meet the growing electricity demand despite the expansion of renewable technologies, resulting in relatively high thermal-based carbon emissions. However, adding bioenergy generation (and transmission lines) helps displace thermal power and decreases emissions while increasing equity in service provision in the Ghanaian case. Bioenergy, which uses agricultural crop and forestry residues and municipal waste as an input⁴⁹, is recognised as Ghana's leading spatially distributed and dispatchable renewable generation technology. Crop residues alone in Ghana have an estimated bioenergy potential of 75 TJ^{49,55}. Other studies have also highlighted the potential of crop residues for bioenergy in Ghana⁵⁶⁻⁵⁸. However, increasing bioenergy could compete with food production for land resources if only relying on energy crops for bioenergy^{59,60}. Bioenergy production, therefore, could be regulated to prevent a negative impact on food security.

The proposed WEFE infrastructure design approach help analysts and policymakers plan for equitable low-carbon futures. The proposed approach considers the spatial equity of water and energy services alongside emission reduction and other aggregated national infrastructure planning goals to identify efficient equitable climate-compatible intervention portfolios for multisector WEFE systems. Pareto-optimal portfolios (Figs. 3 and 4 for Ghana) help identify infrastructure interventions that reduce service inequities and emissions and improve sectoral complementarities. Stakeholders can deliberate the pros and cons of a large set of efficient intervention options to help find a compromise portfolio that manages trade-offs and leverages synergies across

multiple competing interests. The machine learning technique helps stakeholders understand how different interventions drive equity, emission, and other system performance targets, as shown for Ghana (Fig. 5). This allows choosing portfolios or revising intervention choices and repeating the process if no consensus can be attained. The proposed approach could be adapted for other countries planning an equitable transition to lower-emission energy sources, especially those with service inequities and interdependent WEFE challenges.

This study did not consider the potential effects of climate change and uncertainties with future projections in the equity-informed planning of WEFE systems. Climate change could impact energy generation, the demand for water and energy, and water resources management. These changes and future uncertainties could have repercussions on the equity of providing water and energy services and inter-sectoral conflicts among multiple resource systems. Addressing these limitations in future research would help identify robust infrastructure strategies for adapting to climate change.

We now address the comment on how to consider different types of energy technologies for a more complex case study. The power generation types for Ghana are taken from the Ghana Power System Master Plan¹, and include hydro, solar, combined solar and storage, wind, bioenergy, and thermal. The power system model is an interconnected network of different energy sources, transmission lines, substations, distribution networks, and electricity demand locations. Other energy technologies, such as nuclear, offshore wind, and hydrogen, could be represented as nodes in the network, each with their defined constraints, such as maximum power capacity and associated costs. The power system model uses a direct current optimal power flow linear programme to minimise power generation costs at each simulation time step using equality (power balance at each node) and inequality constraints (power generation and transmission line limits). We have revised the 'Power system simulator' section to further describe the components of the power system simulation model. The revised text is in lines 448 to 475 and reads:

The power system model is an interconnected network of different energy sources, transmission lines, substations, distribution networks, and electricity demand locations. Energy generators can be represented as nodes in the network, each with their defined constraints, such as maximum power capacity and associated costs. The Ghana power system model simulates the existing and planned hydro, solar, combined solar and storage, wind, bioenergy, and thermal generators⁴⁹. Electricity demand projections, generating resources, capital and operating costs of power plants and transmission lines, and hourly profiles of renewables were taken from Ghana's Power System Master Plan⁴⁹. Hourly electricity demand profiles and transmission data were collected from the Ghanaian national grid operator, the Ghana Grid Company. The long-term annual peak load projections for 2030-2040 from the Ghana Power System Master Plan⁴⁹ were used to scale the hourly load profile of 2018 to consider the projected growth in the electricity demand in the country (a twofold increase in the peak demand by 2030 compared to 2018). Capital costs consist of engineering, procurement, construction, start-up, and owner costs (for items like land, cooling infrastructure, administration and associated building, site works, project management, and licenses)⁴⁹. The operating costs of variable renewables (solar and wind) and combined solar and storage are lower than the costs of hydropower, bioenergy, and thermal power plants⁴⁹. The operating costs of thermal generators also differ depending on the technology and fuel types. Hourly power system simulations help capture the variability of intermittent renewables (solar and wind) and energy demand, as well as improve the use of hydropower based on the changing availability of renewables and other energy sources. The power system model dispatches the system's generators based on cost-merit prioritisation where solar, wind, combined solar and storage generators are dispatched first, then hydropower, bioenergy, and lastly, thermal generators. This balances energy production to align with variable energy demand and shares of different energy technologies.

Finally, we respond to the question regarding how to deal with high optimisation dimensionality for large, detailed case studies. In this case study, the MOEA optimises 170 decision variables (lines 229 to 233) while simultaneously considering eight equity, environmental, and other system performance objectives. Multi-

objective optimisation search for large, detailed case studies implies high dimensionality, resulting in high computational burden. We are currently addressing this issue in other case studies our research group is working on. Although it is not directly relevant to this paper's scope, we are happy to inform the reviewer that in other (larger) applications we have learned to manage the increased computational burden associated with the increased optimisation dimensionality using machine learning emulators (of conventional simulation codes) and/or using graphics processing units (GPUs) within high-performance parallel computers.

In our response to this comment, we have updated the relevant text in the revised manuscript in lines 229 to 233, as well as in lines 548 to 559, as follows:

Lines 229-233 in the 'Equitable low-carbon energy transition in an African case' section: The design formulation for Ghana includes minimising the electricity access Gini index, carbon emissions from generation, electricity supply curtailment, irrigation water supply Gini index, power system capital and operating costs, and maximising agriculture yields and flood recession agriculture benefits (eight objectives). The multi-objective search optimises 170 interventions (decisions variables), including infrastructure expansion (solar, wind, combined solar and storage, bioenergy, and transmission lines), operating rules for Akosombo, Bui, and Pwalugu reservoirs, and electricity and irrigation water allocation policies.

Lines 548-559 in the 'Methods' section: The design formulation for Ghana includes eight objectives: minimising the electricity access Gini index (equation 1), minimising the carbon emissions from generation (equation 2), minimising the power system load curtailment (equation 3), minimising the irrigation water supply Gini index (equation 4), maximising the agriculture yields (equation 5), minimising the power system capital costs (equation 6), minimising the power system operating costs (equation 7), and maximising the flood recession agriculture benefits (equation 8). The multi-objective search optimises 170 interventions (decisions variables), including the vector of the power system infrastructure expansion, including solar, wind, combined solar and storage, and bioenergy (biomass and biogas) generators, and transmission lines, the

vector of operating rules parameters (equation 9) for Akosombo, Bui, and Pwalugu reservoirs, and the vector of the electricity and irrigation water allocation priority parameters.

3. What are the innovations in findings and methods of this study compared to the authors' previous study? (Designing diversified renewable energy systems to balance multisector performance).

We thank the reviewer for this question. We build on a water resource and power system simulation and design framework introduced by ref.² and extend it to include equity of water and energy services in order to help plan for equitable climate-compatible interventions in multisector resources systems. The previous study (ref.²) shows how integrating intermittent renewables can increase sub-daily river flow variability and multisector conflicts; thus encouraging diversified infrastructure investments with appropriate management. This proposed approach helps consider equity when lowering Ghana's emissions through power system infrastructure expansion (bioenergy, solar, wind, solar storage, and transmission lines), hydropower reservoir re-operation, and electricity and water allocation priority reformulation. Our findings in the case of Ghana suggest equity could be increased by informed infrastructure planning (i.e., by adding new infrastructure and changing allocation priorities in regions with high existing service disparities).

We have expanded the discussion section to summarise the innovation and contribution of this work. The added text also provides further insights into the implications and limitations of the work. The new/revised text, which is in lines 366 to 416 in the discussion section, is provided in the response above addressing the comment 2. We have also summarised the innovation of our study, as follows (in the 'Equity as a policy goal in low-carbon multisector infrastructure planning' section, lines 129 to 138):

Classically, energy infrastructure planning considers goals such as minimising electricity supply curtailment, capital investment, and carbon emissions^{40,41}. Other approaches extend the goals to include water resources and ecosystems, such as maximising hydropower generation and minimising

damage to river ecosystems^{42,43}. While such methods can design infrastructure plans that improve system performance at an aggregate level, equity in the regional distribution of water and energy services has not been considered to our knowledge. In this paper, we build on a water resource and power system simulation and design framework introduced by ref.⁴² and extend it to include spatial equity of water and energy services as infrastructure planning goals.

4. How to select the equitable transition plans? Does this question need methods of game theory to achieve equity?

We thank the reviewer for raising this point. The multi-objective search generates a set of the Pareto-efficient intervention decisions (portfolios in investments and policies) and the energy transition metrics they achieve. The results in our case contain thousands of equally 'optimal' alternatives. These potential solutions can support stakeholder screening and deliberation when seeking to reduce service inequities and emissions and better balance sectoral complementarities.

The reviewer brings up an interesting point; how should stakeholders coalesce on one or a small number of efficient solutions?, and are game theory concepts needed to select or achieve equitable negotiated solutions? In our view, stakeholders can and will negotiate a compromise portfolio that balances trade-offs and leverages synergies across their multiple competing interests. With regard to which solution(s) they will or should select?, here we remain silent. There is 50+ years of literature in game theory and related fields on how negotiating parties make initial decisions, and the dynamics of negotiations; and we view this large and interesting topic as out of scope in the context of our study. The initial solutions individuals or groups of stakeholders will select, and how they will negotiate is hard to characterise and predict, and will vary depending on their institutions, geography, history, economic, political economy, but also on who is in the negotiation room. From our point of view, the proposed method offers a set of equally optimal (Pareto-optimal) solutions, and decision-makers and/or stakeholders must take it from there and make their selection. The extent to which a selected alternative is equitable is up to them.

In our response to this comment and also to a related comment from Reviewer 1, we have added the new text in the revised manuscript in the methods section in lines 509 to 520, as follows:

Pareto-optimal solutions are efficient because they provide the best-achievable compromise among multiple performance objectives that coexist in multi-objective problems. These solutions are non-dominated, indicating no other solutions can outperform or equal them in all objectives at the same time. Stakeholders can consider these optimised solutions to find a compromise that efficiently balances trade-offs among multiple competing interests and leverages synergies. After the search, we used a machine learning method (details provided below) to assess the effectiveness of interventions in achieving performance objectives. Machine learning and interactive data visualisation tools help stakeholders understand, evaluate, and choose a preferred portfolio or iteratively revise the intervention options and repeat the process to obtain alternative, potentially better, intervention portfolios.

5. How the study considers the energy prices under different portfolios?

We thank the reviewer for the comment. The power system model contains operating and capital costs of power plants and transmission lines obtained from the Ghana Power System Master Plan¹. Capital costs consists of engineering, procurement, construction, start-up, and owner costs (for items like land, cooling infrastructure, administration and associated building, site works, project management, and licenses)¹.

The operating costs of variable renewables (solar and wind) and combined solar and storage are lower than the costs of hydropower, bioenergy, and thermal power plants¹. The operating costs of thermal generators also differ depending on the technology and fuel types. We did not consider the energy prices that consumers need to pay, rather we seek to help decision-makers select their power mix based on case-study country operating and capital costs.

Based on this and other related comments, we have revised the ‘Power system simulator’ section to further describe the components of the power system simulation model. The revised text is in lines 448 to 475 and is provided in our response to the comment 2 above.

6. Why use the Random Forest Regression and SHAP method? Is it possible to directly analyze the influence of intervention decisions through the optimization model?

We thank the reviewer for this question. We used the Random Forest Regression models and the Shapley additive explanations (SHAP) to assess the effectiveness of different intervention options (infrastructure assets and resource allocation priorities) in influencing equity, emissions output, and other planning goals. The MOEA typically yields a large set (up to thousands) of optimal solutions in complex optimisation problems such as our Ghana case study, which involves 170 intervention decision variables and eight performance objectives. While common patterns can potentially be identified from directly exploring Pareto-optimal solutions or via cluster analysis, the SHAP of tree-based machine learning models^{3,4} helps explain the relationships between intervention decisions and the MOEA optimised performance measures (the objectives output by the simulators). It provides the importance ranking of different interventions and also the direction and magnitude of their influence in achieving performance objectives. This, along with interactive data visualisation tools (like the <https://www.polyvis.org/> website that makes interactive parallel coordinates plots), can help stakeholders understand, evaluate, and select from amongst efficient solutions.

We have revised the ‘Simulation-based artificial intelligence-assisted multisector system design and machine learning’ section for the use of machine learning models combined with the SHAP in our proposed approach in lines 650 to 663 and in lines 515 to 520, as follows:

Lines 650-663: We used the Random Forest Regression machine learning model⁷⁶ from the Scikit-learn Python library⁷⁷ and the SHAP^{48,78,79} to determine the influence of intervention decisions on equity and other water-energy system

performance indicators. The MOEA typically yields a large set (up to thousands) of optimal solutions for large problems like the Ghana case study. While common patterns can potentially be identified from directly exploring Pareto-optimal solutions or via cluster analysis, the SHAP analysis of tree-based machine learning models^{48,78} helps explain the relationships between intervention decisions and the MOEA performance measures. It provides the importance ranking of different interventions and also the direction and magnitude of their influence in achieving performance objectives. The outcomes of the MOEA are used to train a Random Forest model for each performance objective. The features of each Random Forest model are the decision variables representing different investment and resource allocation priority options, and the target is each of the optimisation objectives.

Lines 515-520: After the search, we used a machine learning method (details provided below) to assess the effectiveness of interventions in achieving performance objectives. Machine learning and interactive data visualisation tools help stakeholders understand, evaluate, and choose a preferred portfolio or iteratively revise the intervention options and repeat the process to obtain alternative, potentially better, intervention portfolios.

7. Line 156-160, this sentence is too long.

We thank the reviewer for the comment. The updated sentence is in lines 155 to 161 and reads:

Intervention options that can be used to achieve the identified equity, environmental, and system performance goals are also selected in this first component of the framework. These interventions include infrastructure assets and resource allocation policies. The selected interventions are used in the second component of the framework to generate a set of efficient multisector infrastructure and operation portfolios using a simulation-based artificial intelligence search design process.

Reviewer 3

The manuscript presents a compelling argument for considering spatial equity in water and energy service provision while planning low-carbon energy transitions, especially in developing countries with multi-sector interdependencies and regional disparities. The proposed analytical framework combining integrated river basin-power system simulation with artificial intelligence design tools is innovative and well-described. The case study of Ghana demonstrates the utility of the framework and the trade-offs involved in achieving equitable low-carbon futures.

The manuscript addresses an important issue in sustainable development. I do have one major comment that requires some additional clarifications, expansions, and refinements, and a couple of minor comments. However, overall, I believe the paper could make a valuable contribution to the literature on equitable low-carbon energy transitions and multi-sector infrastructure planning. Below are my comments:

Major:

- Food: While the paper outlines objectives related to maximizing agricultural yield and maximizing the flood recession agricultural benefit (Equations 5 and 8), the discussion around the direct implications of the infrastructure changes on food systems remains somewhat underdeveloped. Particularly, the interactions between energy infrastructure, specifically bioenergy, and food systems, need deeper exploration.

In lines 322-323 & 372-374, the manuscript identifies investing in bioenergy plants in the Northeast and Northwest regions of Ghana as an effective strategy for improving equity in electricity access. However, it's crucial to consider the broader implications of such investments. Since agriculture is mentioned as an input, research indicates that bioenergy production can compete with food production for land resources, potentially increasing food prices if food crops are used as inputs for bioenergy (see <https://link.springer.com/article/10.1007/s10584-020-02838-8> [link.springer.com] and <https://www.sciencedirect.com/science/article/abs/pii/S0360544210002896?via=ihub> [sciencedirect.com]).

This competition can exacerbate food security challenges, particularly in regions already vulnerable to food access inequalities. The authors could consider expanding the discussion on potential trade-offs between increased bioenergy production and food availability, an essential component of the water-energy-food nexus and equity implications.

We thank the reviewer for this comment. We have expanded the discussion on increasing bioenergy in Ghana for equity in electricity and water services, emissions, load curtailment, and other system performance. The energy infrastructure expansion included in the equity-informed design for the Ghanaian case study are based on Ghana's Power System Master Plan¹. In that plan, bioenergy is recognised as Ghana's main spatially distributed, dispatchable renewable technology, and it is mainly available from agricultural crop and forestry residues and municipal waste^{1,5-8}, with higher potential in the north than other regions (Fig. 2). Increased bioenergy production, however, could negatively affect food security if energy crops are used as an input for bioenergy^{9,10}. The new text in the discussion is in lines 382 to 389 and reads:

Bioenergy, which uses agricultural crop and forestry residues and municipal waste as an input⁴⁹, is recognised as Ghana's leading spatially distributed and dispatchable renewable generation technology. Crop residues alone in Ghana have an estimated bioenergy potential of 75 TJ^{49,55}. Other studies have also highlighted the potential of crop residues for bioenergy in Ghana⁵⁶⁻⁵⁸. However, increasing bioenergy could compete with food production for land resources if only relying on energy crops for bioenergy^{59,60}. Bioenergy production, therefore, could be regulated to prevent a negative impact on food security.

Minor:

1. Line 81. The International Renewable Energy Agency (IRENA) abbreviation is incorrect: "IREA".

We have revised this abbreviation. We thank the reviewer for flagging it.

2. Figures 2, 3, and 4, the captions are in bold, which is effective and clear. However, the descriptions are formatted in normal text that matches the font of the body text. This can lead to confusion, making it difficult for readers to discern where the figure description ends and the main body text resumes.

We thank the reviewer for the feedback. We have made the changes to clearly delineate the figure descriptions from the main body text.

3. Consider expanding the discussion section to further elaborate on the implications of their findings for policy-making and potential limitations. For example, the authors could elaborate on the implications of the findings for policymakers and practitioners, i.e., a detailed discussion of how this research could inform infrastructure planning, policy-making, and sustainable development goals in Ghana and how it could be adapted for other regions or different scales of analysis. Potential limitations, i.e., Uncertainties associated with future projections, climate change impacts, etc.

We thank the reviewer for the comment. We have expanded the discussion section to provide further insights into the implications and limitations of the study. The new/revised text also briefly summarises the innovation and contribution of our work, as suggested by Reviewer 1. This new/revised text is lines 366 to 416; it reads:

Our approach aims to lower national emissions in Ghana through a balanced energy infrastructure system expansion programme (bioenergy, solar, wind, solar storage, and transmission lines), and reservoir re-operation (of Akosombo, Bui, and Pwalugu dams). Ghana's resulting efficient and equitable low-carbon intervention strategies show how considering equity in energy transitions allows reducing both regional service disparities and energy-related greenhouse gas emissions. We show how the spatial distribution of investments drives electricity access equity and that increased investments in renewables and transmission lines, hydropower re-operation, and better spatial planning enhance equity and reduce carbon emissions while improving food production and meeting future energy service goals. With low renewable investments, equity can also be improved, but at the cost of higher emissions and load curtailments. Ghana's power system continues to partially rely on

thermal power plants to meet the growing electricity demand despite the expansion of renewable technologies, resulting in relatively high thermal-based carbon emissions. However, adding bioenergy generation (and transmission lines) helps displace thermal power and decreases emissions while increasing equity in service provision in the Ghanaian case. Bioenergy, which uses agricultural crop and forestry residues and municipal waste as an input⁴⁹, is recognised as Ghana's leading spatially distributed and dispatchable renewable generation technology. Crop residues alone in Ghana have an estimated bioenergy potential of 75 TJ^{49,55}. Other studies have also highlighted the potential of crop residues for bioenergy in Ghana^{56–58}. However, increasing bioenergy could compete with food production for land resources if only relying on energy crops for bioenergy^{59,60}. Bioenergy production, therefore, could be regulated to prevent a negative impact on food security.

The proposed WEFE infrastructure design approach help analysts and policymakers plan for equitable low-carbon futures. The proposed approach considers the spatial equity of water and energy services alongside emission reduction and other aggregated national infrastructure planning goals to identify efficient equitable climate-compatible intervention portfolios for multisector WEFE systems. Pareto-optimal portfolios (Figs. 3 and 4 for Ghana) help identify infrastructure interventions that reduce service inequities and emissions and improve sectoral complementarities. Stakeholders can deliberate the pros and cons of a large set of efficient intervention options to help find a compromise portfolio that manages trade-offs and leverages synergies across multiple competing interests. The machine learning technique helps stakeholders understand how different interventions drive equity, emission, and other system performance targets, as shown for Ghana (Fig. 5). This allows choosing portfolios or revising intervention choices and repeating the process if no consensus can be attained. The proposed approach could be adapted for other countries planning an equitable transition to lower-emission energy sources, especially those with service inequities and interdependent WEFE challenges.

This study did not consider the potential effects of climate change and uncertainties with future projections in the equity-informed planning of WEFE systems. Climate change could impact energy generation, the demand for water and energy, and water resources management. These changes and future uncertainties could have repercussions on the equity of providing water and energy services and inter-sectoral conflicts among multiple resource systems. Addressing these limitations in future research would help identify robust infrastructure strategies for adapting to climate change.

4. Equation 5 could benefit from additional explanations or references to clarify its components and the rationale behind their formulation.

We appreciate the reviewer's comment. Irrigation yields are estimated based on the Food and Agriculture Organization (FAO) Crop Water Requirements method, which is also called the FAO56 Penman-Monteith method¹¹. We have added further details about how irrigation yields are calculated using this FAO method. The updated text, along with equation 5, is in lines 587 to 606 in the methods section and given below.

$$Y = \sum_t \sum_n CR_{t,n} \times (A_n \times y_n) / T \quad (5a)$$

$$CR_{t,n} = r_{t,n} / iwr_{t,n} \quad (5b)$$

$$iwr_{t,n} = \sum_{ct \in n} cwr_{t,(ct \in n)} / (\alpha_{ct} \times \beta_{ct}) \quad (5c)$$

$$cwr_{t,(ct \in n)} = \max(0, (Kc_{t,(ct \in n)} \times ET o_{t,(ct \in n)} - R_{t,n}) \times A_{(ct \in n)}) \quad (5d)$$

where, Y is the total irrigation yield in tonnes per year estimated using the Food and Agriculture Organization (FAO) Crop Water Requirements method⁶⁵, $CR_{t,n}$ is the irrigation water supply curtailment ratio, A_n is the area in ha per irrigation scheme n , y_n is the annual yield in tonnes per ha per irrigation scheme n , t is the simulation time step, T is the number of simulation years, $r_{t,n}$ is the crop water allocated by the model, $iwr_{t,n}$ is the irrigation water requirement for irrigation scheme n , $cwr_{t,n}$ is the crop water requirement per irrigation scheme n , α_{ct} is the application efficiency (assumed 80%), β_{ct} is conveyance efficiency (assumed 70%), and $Kc_{t,(ct \in n)}$, $ET o_{t,(ct \in n)}$, and $R_{t,n}$ are crop water coefficients,

reference evapotranspiration in mm per day, and effective rainfall in mm per day obtained from ref. ⁶⁶.

R , K_c , and ET_o are used in the FAO method⁶⁵ to calculate the water requirement for each crop. ET_o is the evapotranspiration of a standardised reference crop, while K_c is the factor to change ET_o for specific crop water requirements. R is compared with the amount of water a crop needs ($K_c \times ET_o$) to determine whether irrigation is required for that crop. Crops that are considered vary for each irrigation scheme and include rice, maize, sugar cane, beans, tomatoes, and fresh vegetables. The total irrigation water requirement for each irrigation scheme is calculated by assuming the overall irrigation efficiency for surface irrigation. The level of irrigation water supply compared to the water required is then used to estimate the total irrigation yield.

References

1. USAID. Integrated Power System Master Plan for Ghana. (2018).
2. Gonzalez, J. M. *et al.* Designing diversified renewable energy systems to balance multisector performance. *Nat. Sustain.* 1–13 (2023).
3. Lundberg, S. M. *et al.* From local explanations to global understanding with explainable AI for trees. *Nat. Mach. Intell.* **2**, 56–67 (2020).
4. Lundberg, S. M. & Lee, S.-I. A unified approach to interpreting model predictions. *Adv. Neural Inf. Process. Syst.* **30**, (2017).
5. Nelson, N. *et al.* Potential of bioenergy in rural Ghana. *Sustainability* **13**, 381 (2021).
6. Mohammed, Y. S., Mokhtar, A. S., Bashir, N. & Saidur, R. An overview of agricultural biomass for decentralized rural energy in Ghana. *Renew. Sustain. Energy Rev.* **20**, 15–25 (2013).
7. Duku, M. H., Gu, S. & Hagan, E. B. A comprehensive review of biomass resources and biofuels potential in Ghana. *Renew. Sustain. Energy Rev.* **15**, 404–415 (2011).
8. Azasi, V. D., Offei, F., Kemausuor, F. & Akpalu, L. Bioenergy from crop residues: A regional analysis for heat and electricity applications in Ghana. *Biomass and Bioenergy* **140**, 105640 (2020).
9. Hasegawa, T. *et al.* Food security under high bioenergy demand toward long-term climate goals. *Clim. Change* **163**, 1587–1601 (2020).
10. Brinkman, M. *et al.* The distribution of food security impacts of biofuels, a Ghana case study. *Biomass and Bioenergy* **141**, 105695 (2020).
11. Allen, R. G., Pereira, L. S., Raes, D. & Smith, M. Crop evapotranspiration - Guidelines for computing crop water requirements. (1998).

Response to the reviewers' comments

Below are our responses (blue text) to reviewer comments (black text).

Reviewer 1

The authors of this article have made good revisions or supplemented explanations one by one based on the comments. I recommend publishing this article if the code is feasible.

We appreciate the reviewer's feedback and recommendation for publication. We have included data and code availability statements in the manuscript. We have provided links to the codes and data that are freely available. The data for the river system model can be made available upon presentation of the necessary permission from the Ghana Water Research Institute that owns the data. Source data are provided for the figures.

Unable to access this page (<https://github.com/pywr/pywr>).

Thank you. We confirm that the Python Water Resources (Pywr) GitHub repository is accessible at: <https://github.com/pywr/pywr>. The Pywr, Pyenr, and Pynsim software libraries used to develop the river basin-power system model are general-purpose, open-source, and freely available on GitHub.

Reviewer 3

I have no more comments. All my comments have been addressed to my satisfaction.

Congratulations to the authors.

We thank the reviewer and appreciate their feedback.

There are two distinct codebases in this project: one for the simulation and another for postprocessing. While I have not personally installed or tested either codebase, both are open-source. The postprocessing codes utilize general purpose, open-source libraries that are freely available to everyone.

We thank the reviewer the comment. The Pywr, Pyenr, and Pynsim software libraries used to develop the river basin-power system model are general-purpose, open-source, and freely available on GitHub. The software libraries used for machine learning and for assessing the effectiveness of interventions in achieving performance objectives are also open-source and freely available.

Comments

The consideration of low-carbon and equity multisector infrastructure planning is a hot and difficult issue in the world. This study proposes an analytical framework, taking Ghana as the case study area, considering the regional distribution of basic services such as water and energy, and combining comprehensive simulation of watershed power systems with artificial intelligence design tools. By determining effective infrastructure intervention combinations and their implicit trade-offs between spatial equity, carbon emissions, food production, and river ecosystem performance in providing water and energy services, demonstrate the utility of this framework for Ghana. This study has important theoretical and practical significance for achieving carbon neutrality goals in the fields of water resources and energy. I suggest making major revisions before accepting. The following opinions and suggestions are for reference:

(1) The main text only uses 'Ghana' as a case study area for application analysis. It is recommended to add relevant content such as 'Ghana' or 'Ghana as a case study area' to the title.

(2) In the section on 'Equity as a policy goal in low-carbon multisector infrastructure planning', it is pointed out that 'multisector infrastructure' is somewhat fuzzy in the main text; Additional clarification is needed on the specific departments involved. If there are many departments, they can be listed in the relevant table as an attachment.

(3) Lines 36-37, in the abstract section, which points out that 'The utility of the framework is demonstrated for Ghana by identifying the most efficient infrastructure intervention portfolios and', but this section in the main text is somewhat fuzzy; Further explanation is needed to determine the key process for determining the 'most effective' reasoning; Provide sufficient detailed explanations.

(4) Lines 443-444, the article points out that 'A maximum of 1.5 million iterations is specified as a stopping criterion in the design process', and several key iteration processes need to be provided, along with clear explanations of the corresponding algorithms; Please provide key details.

(5) Suggest checking of the figures appearing in the text and supplementary materials, and making modifications to them, such as Fig. S2, the differentiation between the vertical axis curves is not high, and it needs to be redrawn more clearly.

(6) It is recommended to conduct a comprehensive check on the citation of references in the text. For example, the article points out that MOEA uses references [55-58] to simulate the natural biological evolution process, and later uses references [59, 60, 60, 61, 62]. There are too many references used, and the evolutionary algorithm used in this study may only have 1-3 references. Please explain clearly; It is necessary to write how this method was applied to this study, as the text is somewhat vague; Please remove any unused references.

(7) Lines 312-315, in Figure 4, 'The figure also shows that there is more infrastructure expansion in the Northwest and Northeast regions across the selected portfolios and that the intra-regional transmission line expansion (sum of first six coloured bars in a2-

f2) is higher than cross-regional transmission line expansion (sum of last six coloured bars in a2-f2)'. It is difficult to see that the expansion of transmission lines within the region is higher than that of cross regional transmission lines. Please further explain.

(8) Although this article argues hot research questions, the work involves multiple departments and fields, which is quite complex. It is recommended to further summarize the innovation and contribution of this article in the discussion section.